# Break Stylistic Sophon: Are We Really Meant to Confine the Imagination in Style Transfer?

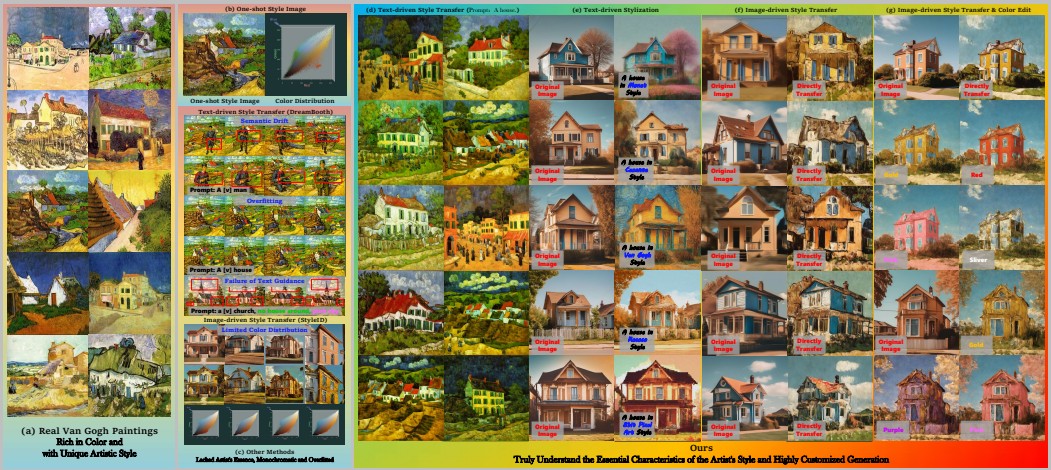

Figure 1: We found that existing image style transfer methods based on a single style image (b) either suffer from overfitting and semantic drift when performing text-driven style transfer or merely achieve texture blending rather than truly learning the artist's style during image-driven style transfer (c). Moreover, there is currently a lack of a unified framework for addressing the various issues in style transfer. For genuine artworks (a), the colors are never confined to just one piece (b). Therefore, this paper designs a unified model that can learn the artist's creative style and achieves style transfer results that are indistinguishable from the artist's creative style (d), (e) and (f) through different pipeline. Meanwhile, the model also realizes color editing during the style transfer process for the first time (g).

## Abstract

In the realm of image style transfer, existing algorithms relying on single reference style images encounter formidable challenges, such as severe semantic drift, overfitting, color limitations, and a lack of a unified framework. These issues impede the generation of high quality, diverse, and semantically accurate images. In this study, we introduce *StyleWallfacer*, an innovative unified training and inference framework, which not only addresses various issues encountered in the style transfer process of traditional methods but also unifies the framework for different tasks. This framework is designed to advance the development of this field by enabling high-quality style transfer and text driven stylization. First, we propose a semantic-based style injection method that uses BLIP to generate text descriptions strictly aligned with the semantics of the style image in CLIP space. By leveraging a large language model to remove style-related descriptions from these descriptions, we create a semantic gap. This gap is then used to fine-tune the model, enabling efficient and drift-free injection of style knowledge. Second, we propose a data augmentation strategy based on human feedback, incorporating high-quality samples generated early in the fine-tuning process into the training set to facilitate progressive learning and significantly reduce its overfitting. Finally, we design a training-free triple diffusion process using the fine-tuned model, which

manipulates the features of self-attention layers in a manner similar to the cross-attention mechanism. Specifically, in the generation process, the key and value of the content-related process are replaced with those of the style-related process to inject style while maintaining text control over the model. We also introduce query preservation to mitigate disruptions to the original content. Under such a design, we have achieved high-quality image-driven style transfer and text-driven stylization, delivering high-quality style transfer results while preserving the original image content. Moreover, we achieve image color editing during the style transfer process for the first time, further pushing the boundaries of controllable image generation and editing technologies and breaking the limitations imposed by reference images on style transfer. Our experimental results demonstrate that our proposed method outperforms state-of-the-art methods.

# 1 INTRODUCTION

Art encapsulates human civilization's essence, epitomizing our imagination and creativity, and has yielded innumerable masterpieces. Online, you may encounter a painting that profoundly affects you, yet you may find it hard to describe the artist's unique style or locate more similar works. This highlights a key issue in image generation: style transfer.

Recently, numerous excellent works have conducted research on this issue, which are mainly divided into three categories: text-driven style transfer Rout et al. (2025); Sohn et al. (2023a), image-driven style transfer Chung et al. (2024); Liu et al. (2021); Huang & Belongie (2017) and text-driven stylization Jiang & Chen (2024); Brooks et al. (2023); Tumanyan et al. (2023). The mainstream approach of image-driven style transfer is to decouple the style and content information of a reference style image, and then inject the style information as an additional condition into the model's generation process Chen et al. (2024); Wang et al. (2024c). This enables the model to generate new content that is similar to the reference style image in terms of texture and color. Alternatively, a unique identifier can be used to characterize the style of the style image, and the model can be fine-tuned to learn new stylistic knowledge for text-driven style transfer Ruiz et al. (2023); Han et al. (2023). This allows the model to recognize and generate corresponding style images using the identifier. For text-driven stylization, most methods involve blending the pre-trained model's prior style knowledge with the texture of the target image to achieve the final style transfer result Jiang & Chen (2024). However, as shown in Figure 1 (c), these models generally suffer from the following issues:

**Limited color domain**: Both the style-content disentanglement-based and identifier-based fine-tuning approaches commonly face the problem of a restricted color domain in the generated images. Specifically, the color distribution of the generated images is entirely consistent with that of the single reference style image. For example, in the case of Van Gogh's paintings Ojha et al. (2021b) as shown in Figure 1 (a), great artists are by no means limited to the color palette of a single artwork in Figure 1 (b). Therefore, such generation results are unreasonable. For more detailed visualization results and discussions, please refer to Appendix F.1.

**Failure of text guidance**: Due to the architectural flaws in the style-content disentanglement-based methods and the mismatch between text and image in the style information injection of the identifier-based fine-tuning methods, models exhibit significant semantic drift, which refers to the phenomenon of inconsistency or deviation in semantics between the generated image and the input text prompt in the T2I model. This not only leads to chaotic generation but also results in the loss of the model's ability to handle complex text prompts. For more detailed visualization results and discussions, please refer to Appendix F.2.

**Risk of overfitting** : Due to the extremely limited number of training samples, traditional approaches are generally prone to overfitting. This results in a loss of structural diversity in the generated content. For more detailed visualization results and discussions, please refer to Appendix F.3.

**Lack of a unified framework**: Due to the significant differences between various style transfer tasks, most existing style transfer methods are only capable of handling one specific task, and there is a lack of a unified framework to integrate these tasks.

These problems, much like the "sophons" in "The Three-Body Problem" Liu (2014) that restrict human technological progress, limit people's imagination for style transfer. In fact, truly good style

transfer enables the model to learn and imitate the artist's style, rather than mechanically copying the textures of reference images, thus achieving true artistic creation. However, most of the above-mentioned studies are limited to using a single style image as reference, thus suffering from the aforementioned problems. In few-shot style transfer, however, the number of reference style samples increases; the model is no longer constrained to learning the sole texture present in one sample, but can instead acquire richer stylistic knowledge. Therefore, we aim to bridge research between these two areas, leveraging ideas from few-shot style transfer to assist single-shot style transfer in achieving more realistic stylization results. Just as in "The Three-Body Problem", humanity uses the "Wallfacer Plan" to break the technological blockade imposed by the "sophons", to break the limitations of "sophons" in style transfer, this paper proposes a novel unified style transfer framework, called *StyleWallfacer*, which consists of three main components:

Firstly, a style knowledge injection method based on semantic differences is proposed (Figure 2 (a)). By using BLIP Li et al. (2022) to generate text descriptions that are strictly aligned with the target style image in CLIP space Radford et al. (2021), and then leveraging LLM to remove the style-related descriptions, a semantic gap is created. This gap allows the model to maintain its prior knowledge as much as possible during training, focusing solely on learning the style information. As a result, the model captures the most fundamental stylistic elements of the style image (e.g., the artist's brushstrokes). As shown in Figure 1 (d), this not only enables the generation of new samples with rich and diverse colors but also preserves the model's ability to handle complex text prompts.

Secondly, a progressive learning method based on human feedback (HF) is employed (Figure 5). At the beginning of model training, the model is trained using a single sample. During the training process, users are allowed to select high-quality samples generated by the model and add them to the training set. This effectively expands the single-sample dataset and significantly mitigates overfitting of the model.

Thirdly, we propose a brand-new training-free triple diffusion "style-structure" diffusion process (Figure 2 (b) and (b1)). It explores the impact of different noise thresholds on the model's generation effects by using the diffusion process with a smaller noise threshold as the main process to preserve the content information of the original image, and employing the diffusion process with a larger noise threshold as the style guidance process. Meanwhile, the *Key* and *Value* from the self-attention layer during this process are extracted to replace the *Key* and *Value* in the main diffusion process and obtain the initial noise of the style image to be transferred through DDIM inversion Song et al. (2021). The *Query* from the diffusion process of the inverted noise is extracted and fused with the *Query* in the main diffusion process, serving as a structural guidance for the main diffusion process. Meanwhile, the pre-trained style LoRA Hu et al. (2022) is used as a style guide to direct the model to conduct image-driven style transfer. This approach thus achieves high-quality style transfer results as shown in Figure 1 (e) and (f). During the generation process, text prompt is employed as a condition, and in combination with the aforementioned structure, it also enables color editing of the model during the style transfer process as shown in Figure 1 (g).

Our main contributions are summarized as follows:

**(1)** We propose the first unified style transfer framework that simultaneously achieves high-quality style transfer from the perspective of the task. Meanwhile, for the first time, it enables text-based color editing during the style transfer process.

**(2)** We propose a style knowledge injection method based on semantic differences, which achieves efficient style knowledge injection without affecting the model's semantic space and suppresses semantic confusion during the style injection process.

**(3)** We propose a progressive learning method based on human feedback for few-shot datasets, which alleviates the model overfitting caused by insufficient data and significantly improves the generation quality after model training.

**(4)** We propose a novel training-free triple diffusion process that achieves high-quality style transfer results while retaining the control ability of text prompts over the generation results, and for the first time enables color editing during the style transfer process.

**(5)** Our experiments demonstrate that the proposed method in this paper addresses many issues encountered by traditional methods during style transfer, achieving high-quality style generation results rather than merely texture blending, and delivering state-of-the-art performance.

## 2 STYLEWALLFACER

### 2.1 OVERALL ARCHITECTURE OF *StyleWallfacer*

As shown in Figure 2, *StyleWallfacer* mainly consists of two parts: First is the semantic-based style learning strategy, which aims to guide the model to learn the most essential style features in artworks based on the semantic differences between images and their text descriptions during the model fine-tuning process, truly helping the model understand the artist's style. It also employs a data augmentation method based on human feedback to suppress overfitting when the model is fine-tuned on a single image, thereby achieving realistic text-driven style transfer.

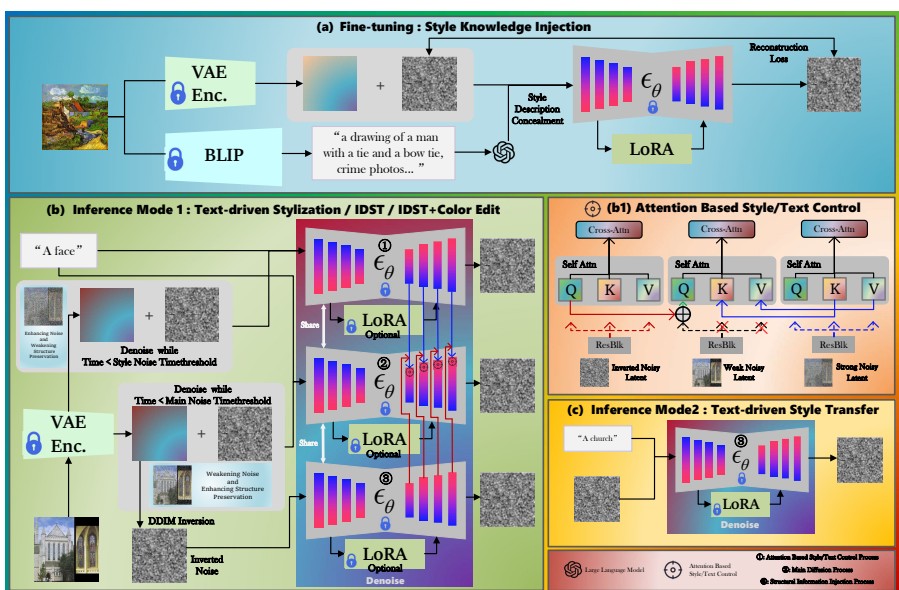

Figure 2: **Illustration of the *StyleWallfacer* Framework.** In the fine-tuning stage (a), we use a semantic-based style knowledge injection method with human feedback (see Figure 5) fine-tuning to help the model learn the style knowledge of a single image, obtaining fine-tuned style LoRA weights. This enables powerful text-driven style transfer (c). In the inference stage (b), we design a triple training-free diffusion pipeline (denoted as ①, ②, ③). It use the diffusion denoising process with a smaller threshold $t_s^s$ as the main process and extract *Key* $\mathbf{K}_t^l$ and *Value* $\mathbf{V}_t^l$ from the process with a larger threshold $t_s^l$ to guide the main process in style and text. Additionally, we use the DDIM inversion latent's denoising diffusion process as the third guiding process, extracting its *Query* $\mathbf{Q}_t^i$ to inject into the main process, achieving high-quality image-driven style transfer, text-driven stylization and color edit (b1). For more detailed introductions to the pipelines, please refer to Appendix B.3.

The second part is the training-free triple diffusion process, which is designed using the previously fine-tuned LoRA weights. This section comprises three newly designed pipelines tailored to address different style transfer problems. By adjusting the self-attention layers of three denoising networks that share weights (denoted as ⊕), it achieves high-quality style control and, for the first time, enables text prompts to control image colors during the style transfer process, solving the traditional method's shortcomings of monochromatic colors, simple textures, and lack of text control when transferring styles based on a single image.

### 2.2 SEMANTIC-BASED STYLE LEARNING STRATEGY

The semantic-based style learning strategy primarily aims to fine-tune text-to-image (T2I) models using their native "language" to enhance their comprehension of the knowledge humans intend them to learn during the fine-tuning process. Taking Stable Diffusion as an example, there is a significant discrepancy between the image semantics understood by the pre-trained CLIP and the intuitive human understanding of image semantics. Therefore, to better "communicate" with the pre-trained T2I model during fine-tuning, this paper employs a method of reverse-engineering the semantic

information of image $\mathbf{I}$ in the CLIP space through BLIP Li et al. (2022):

$$\mathbf{T}_{\text{CLIP}} = \text{BLIP}\,(\mathbf{I}) \tag{1}$$

where $\mathbf{T}_{\text{CLIP}}$ denotes the image prompt derived through BLIP.

Although such methods enable us to obtain the semantic information corresponding to an image in the CLIP space, this text description cannot be directly employed in the fine-tuning process. This is because the description $\mathbf{T}_{\text{CLIP}}$ encompasses all information pertaining to the image, including content, style, and other details understood by CLIP. Utilizing this comprehensive description for fine-tuning still results in the model's inability to comprehend human fine-tuning intentions, thereby preventing it from learning the stylistic information of the dataset.

Therefore, our *StyleWallfacer* transforms $\mathbf{T}_{\text{CLIP}}$ by creating an semantic discrepancy among descriptions. By incorporating a large language model to perform subtle semantic edits on $\mathbf{T}_{\text{CLIP}}$, descriptions related to image style are selectively removed:

$$\mathbf{T}_{w/oS} = \text{LLM}\,(\mathbf{T}_{\text{CLIP}}) \tag{2}$$

where $\mathbf{T}_{w/oS}$ denotes the text description after removing the style information, and LLM stands for large language model.

After such processing, we obtain the image $\mathbf{I}$ and its corresponding text description $\mathbf{T}_{w/oS}$ in the CLIP space, from which stylistic descriptions have been removed. As shown in Figure 2 (a), fine-tuning a pre-trained T2I model using these image-text pairs enables it to focus more effectively on understanding stylistic information, thereby circumventing unnecessary semantic drift.

### 2.3 TRAINING-FREE TRIPLE DIFFUSION PROCESS

After fine-tuning, the model has essentially learned the most fundamental style knowledge from the reference style image. Therefore, how to activate this knowledge so that it can be utilized for image-driven style transfer has become an extremely critical issue.

Unlike traditional one-shot style transfer algorithms that require the reference style image as input during style transfer, we aim to rely solely on the pre-trained style LoRA obtained in Section 2.2 for style transfer. Therefore, we cannot adopt a method similar to StyleID Chung et al. (2024) to manipulate the features in the self-attention layer as if they were cross-attention features, with the features from the style image $\mathbf{I}_s$ serving as the condition for style injection.

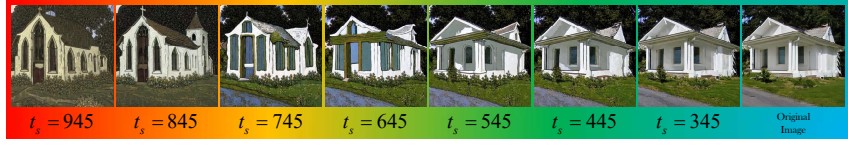

Figure 3: **Illustration of the Impact of Noise Schedule Threshold $t_s$ on Model Generation Results.**

However, as shown in Figure 3, we observe that when initializing the noisy latent $\mathbf{X}_0$ with the original image and using the U-Net to denoise it, the larger the noise schedule threshold $t_s$, the more stylized the generated image will be, losing the original image's content information and retaining only its basic semantics. Conversely, the smaller the noise schedule threshold $t_s$, the more the model's generation tends to preserve the original image's content information, while reducing the diversity and stylization in the generation process.

Therefore, we contemplate: Is it possible to fully leverage this characteristic by employing a diffusion process with a smaller $t_s$ as the main diffusion process, and using a diffusion process with a larger $t_s$ as the stylistic guiding process? Meanwhile, we can utilize the inverted latent obtained through DDIM inversion as the noisy latent for the third diffusion process, and harness the residual information from its denoising process as content guidance. In this way, we aim to achieve high-quality style transfer results while preserving the image content.

To this end, as shown in Figure 2 (b), we first use the VAE encoder to transform the image $\mathbf{I}_c$ to be transferred from the pixel space to the latent space, obtaining $\mathbf{F}_0$. By setting a larger noise schedule

threshold $t_s^l$, we add noise to $\mathbf{F}_0$ (at $t = 0$) to obtain $\mathbf{F}_l$ (at $t = t_s^l$). Similarly, by using a smaller noise schedule threshold $t_s^s$, we obtain $\mathbf{F}_s$ (at $t = t_s^s$). Additionally, we use DDIM inversion to invert $\mathbf{F}_0$ to Gaussian noise $\mathbf{F}_i$ (at $t = T$). Then, using the same denoising U-Net, we denoise $\mathbf{F}_s$, $\mathbf{F}_l$, and $\mathbf{F}_i$ respectively. As shown in Figure 2 (b1), during the entire denoise process of latent $\mathbf{F}_s$, we transfer $\mathbf{F}_s$ to $\mathbf{F}_l$ by injecting the *Key* $\mathbf{K}_t^l$ and *Value* $\mathbf{V}_t^l$ collected from $\mathbf{F}_l$ into the self-attention layer, instead of the original *Key* $\mathbf{K}_t^s$ and *Value* $\mathbf{V}_t^s$. However, merely implementing this substitution can result in content disruption, as the content of the $\mathbf{F}_s$ representation would be progressively altered with the changes in the attended values.

Consequently, we propose a query preservation mechanism to retain the original content. Simply, as shown in Figure 2 (b1), we fuse the *Query* $\mathbf{Q}_t^i$ of DDIM inverted latent $\mathbf{F}_i$ with the original *Query* $\mathbf{Q}_t^s$ to get *Query* $\mathbf{Q}_t^f$ and inject it to the main denoise process instead of the original *Query* $\mathbf{Q}_t^s$. These style injection, query preservation and structural residual injection processes at time step t are expressed as follows:

$$\mathbf{Q}_t^f = \beta\mathbf{Q}_t^i + (1 - \beta)\mathbf{Q}_t^s, \tag{3}$$

$$\phi_{\text{out}}^l = \text{Attn}(\mathbf{Q}_t^f, \mathbf{K}_t^l, \mathbf{V}_t^l), \tag{4}$$

where $\beta \in [0, 1]$. $\mu(\cdot)$, $\sigma(\cdot)$ and $\phi_{\text{out}}^l$ denote channel-wise mean, standard deviation and the result of self-attention calculation after replacement, respectively. In addition, we apply these operations on the decoder of U-net in SD. We also highlight that the proposed method can adjust the degree of style transfer by changing noise schedule threshold $t_s^l$ and $t_s^s$. Specifically, lower $t_s^l$ and $t_s^s$ maintains more content, while higher $t_s^l$ and $t_s^s$ strengthens effects of style transfer.

## 2.4 Data Augmentation for Small Scale Datasets Based on Human Feedback

Although this paper proposes a more robust style knowledge injection method than DreamBooth Ruiz et al. (2023) in Section 2.2, fine-tuning models with a single sample remains challenging. Therefore, inspired by human feedback reinforcement learning (HFRL) Shen et al. (2025), this paper proposes a human feedback-based data augmentation method for small-scale datasets to compensate for dataset insufficiency and mitigate overfitting.

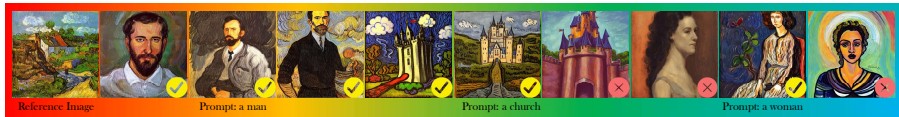

Figure 4: **Illustration of the Best Generation Results When Fine-tuning the Model Directly with a Single Image.**

Specifically, when the model is first trained on a single style image, as shown in Figure 4, although the injection of style knowledge does not generalize well to all the prior knowledge of the model in the early stages of training, and some of the generated results do not match the reference style consistently, resulting in an asynchronous phenomenon in the injection of style knowledge. However, there are still many excellent samples in the model's generated results. The reason for the emergence of these samples is that the prior knowledge represented by these samples is similar to the style image used in training in the CLIP space. This makes it easier for the model to transfer style knowledge to these pieces of knowledge during training. Therefore, it is possible to select samples that meet the style requirements from the large number of generated samples and add them to the training set for further fine-tuning of the model.

To this end, as shown in Figure 5, we divide the model's fine-tuning process into three stages. In the first stage, which is the single-sample fine-tuning stage, we generate a large number of new samples using a text prompt that reflects the basic semantics of the reference image after the model has been trained. We then manually select the 50 samples that best match the stylistic features of the reference image and add them to the training set for the second stage of fine-tuning. In the second stage of fine-tuning, the basic idea is similar to the first stage. We expand the training set from 50 to 100 images. Finally, we fine-tune the model using these 100 samples to obtain the final style LoRA.

Through this data augmentation strategy, the overfitting phenomenon of the model during the fine-tuning process is greatly alleviated. The fine-tuned model is able to generate more diverse results and

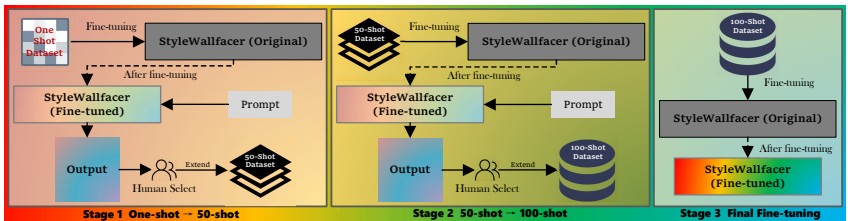

Figure 5: **Illustration of the Small Scale Datasets Augmentation Method Based on Human Feedback.**

generalize the style knowledge to all the prior knowledge of the model, rather than being limited to a single sample.

## 3 EXPERIMENTS

### 3.1 EXPERIMENTAL SETTING

**Baselines** Our baseline list in one-shot text-driven style transfer includes DreamBooth Ruiz et al. (2023), a LoRA Hu et al. (2022) version of DreamBooth, Textual Inversion Gal et al. (2023), and SVDiff Han et al. (2023). For the text-driven stylization task, the baselines we selected include Artist Jiang & Chen (2024), InstructPix2Pix Brooks et al. (2023) , and Plug-and-play (PnP) Tumanyan et al. (2023). And baseline list in one-shot image-driven style transfer includes StyleID Chung et al. (2024), AdaAttn Liu et al. (2021), AdaIN Huang & Belongie (2017), AesPA-Net Hong et al. (2023) and InstantStyle-Plus Wang et al. (2024c).

**Datasets** We selected one image from each of the three widely used 10-shot datasets, including landscapes Wang & Tang (2009), Van Gogh houses Ojha et al. (2021b), and watercolor dogs Sohn et al. (2023b), to form our one-shot datasets, in order to quantitatively evaluate the proposed method from a better perspective. To test our model, we first used FLUX Labs (2024) to generate 1,000 images of houses, 1,000 images of dogs and 1,000 images of mountains based on the prompts "a photo of a house", "a photo of a dog" and "a photo of a moutain", respectively. These images served as the style-free images to be transferred.

**Metric** For image style similarity, we compute CLIP-FID Parmar et al. (2022), CLIP-I score, CLIP-T score and DINO score Zhang et al. (2023a) between 1,000 samples with the full few-shot datasets. For image content similarity, we compute the LPIPS Parmar et al. (2022) between 1,000 samples and the source image to evaluate the content similarity between the style-transferred images and the original images. Intra-clustered LPIPS Ojha et al. (2021a); Zhang et al. (2018) of 1,000 samples is also reported as a standalone diversity metric.

**Detail** For other details of the experiments and *StyleWallfacer*, please refer to Appendix B.

### 3.2 QUALITATIVE COMPARISON

**One-shot Text-driven Style Transfer Experimental Qualitative Results.** As depicted in Figure 6, *StyleWallfacer* outperforms other methods in generating diverse and semantically accurate images. Unlike other methods that suffer from overfitting and semantic drift when trained on single-style images, *StyleWallfacer* employs multi-stage progressive learning with human feedback to reduce overfitting and enhance diversity. It also avoids identifiers for style injection, minimizing semantic drift and enabling precise style generation based on prompts.

**Text-driven Stylization Experimental Qualitative Results.** As shown in Figure 7, other methods, except *StyleWallfacer*, although have completed the task of style transfer, the results obtained after the transfer are far from the authentic style of the painter and fall short of the expected level. However, *StyleWallfacer* has achieved the best balance between image style transfer and content preservation. The images after style transfer not only closely match the painter's authentic style but also feature finer details and a high degree of fidelity to the original image content.

**One-shot Image-driven Style Transfer Experimental Qualitative Results.** As shown in Figure 8

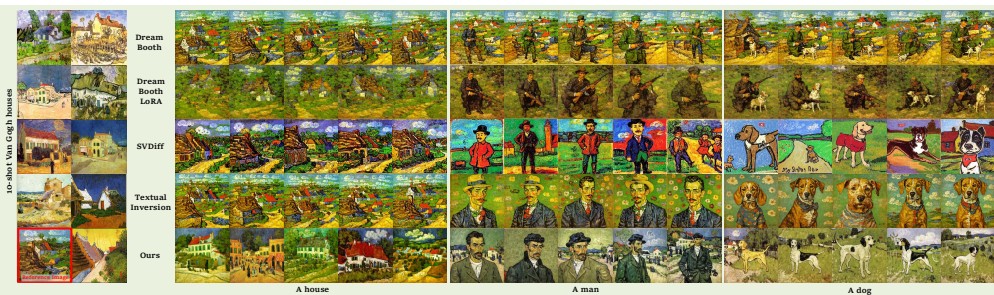

Figure 6: **Qualitative Comparison of Text-driven Style Transfer Results on Van Gogh houses Dataset Using Different Methods.** Due to page limitations, we have placed some of the experimental results in Appendix H.1.

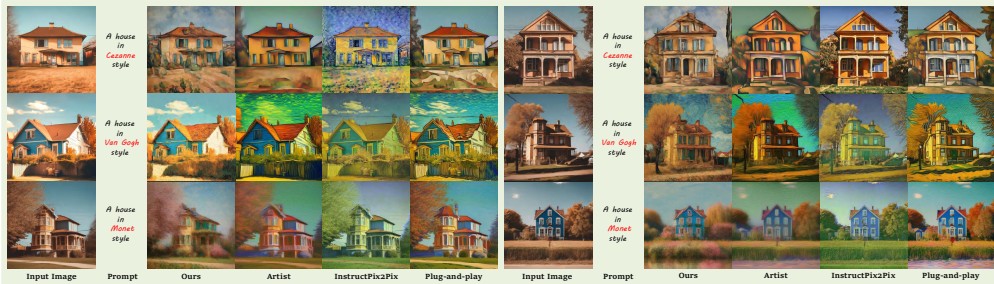

Figure 7: **Qualitative Comparison of Text-driven Stylization Results Using Different Methods.** Due to page limitations, we have placed some of the experimental results in Appendix H.2.

(a), a visual comparison of the style transfer results of the *StyleWallfacer* model with other methods is presented. Clearly, the *StyleWallfacer* model achieves the best results in terms of image structure preservation and style transfer. Compared with the results of other methods, the style transfer results of *StyleWallfacer* have truly realized the style transfer, as if the painter himself had redrawn the original image according to his painting style, rather than simply blending the textures and colors of the original and reference images. Moreover, in terms of detail, the results generated by *StyleWallfacer* feature more refined texture details, while other methods generally suffer from noise and damage.

**One-shot Image-driven Style Transfer & Color Edit Experimental Qualitative Results.** As depicted in Figure 8 (b), the visualization results of image-driven style transfer and color editing are presented. Analysis of the figure reveals that the proposed method in this paper not only accomplishes style transfer but also retains the model's controllability via text prompts. This enables synchronous guidance of the model's generation process by both "text and style", thereby enhancing controllability. Moreover, the images obtained after style transfer maintain a high degree of content consistency with the original images, achieving a better balance between generation diversity and controllability.

## 3.3 QUANTITATIVE COMPARISON

| Method | Landscapes (one-shot) | | | | Van Gogh Houses (one-shot) | | | | Watercolor Dogs (one-shot) | | | |
|---|---|---|---|---|---|---|---|---|---|---|---|---|
| | CLIP-FID ↓ | DINO ↑ | CLIP-I ↑ | I-LPIPS ↑ | CLIP-FID ↓ | DINO ↑ | CLIP-I ↑ | I-LPIPS ↑ | CLIP-FID ↓ | DINO ↑ | CLIP-I ↑ | I-LPIPS ↑ |
| DreamBooth* Ruiz et al. (2023) | 29.25 | 0.8565 | 0.8611 | 0.7878 | 28.95 | 0.8480 | 0.8224 | 0.7553 | 35.31 | 0.8224 | 0.7648 | 0.6570 |
| DreamBooth+LoRA* Hu et al. (2022) | 29.54 | 0.8489 | 0.8628 | 0.7200 | 31.08 | 0.8316 | 0.8000 | 0.6611 | 37.78 | 0.8510 | 0.8124 | 0.7145 |
| SVDiff* Han et al. (2023) | 29.53 | 0.8406 | 0.8648 | 0.7301 | 27.76 | 0.8641 | 0.8642 | 0.7435 | 45.09 | 0.7670 | 0.7854 | 0.6815 |
| Text Inversion* Gal et al. (2023) | 30.58 | 0.8425 | 0.8513 | 0.6947 | 29.35 | 0.8488 | 0.8245 | 0.7616 | 27.77 | 0.8393 | 0.7964 | 0.6941 |
| Ours | 28.34 | 0.8649 | 0.8712 | 0.8388 | 26.44 | 0.8649 | 0.8732 | 0.7084 | 26.64 | 0.8608 | 0.8540 | 0.7205 |

Table 1: **Quantitative Comparisons to SOTAs on Text-driven Style Transfer Task.** The results that achieve the highest and second-highest performance metrics are respectively delineated in red and blue.

As shown in Table 1, Table 2 and Table 3, the method proposed in this paper achieved the best results compared with all the baseline methods, further demonstrating the effectiveness of the proposed method from a quantitative perspective.

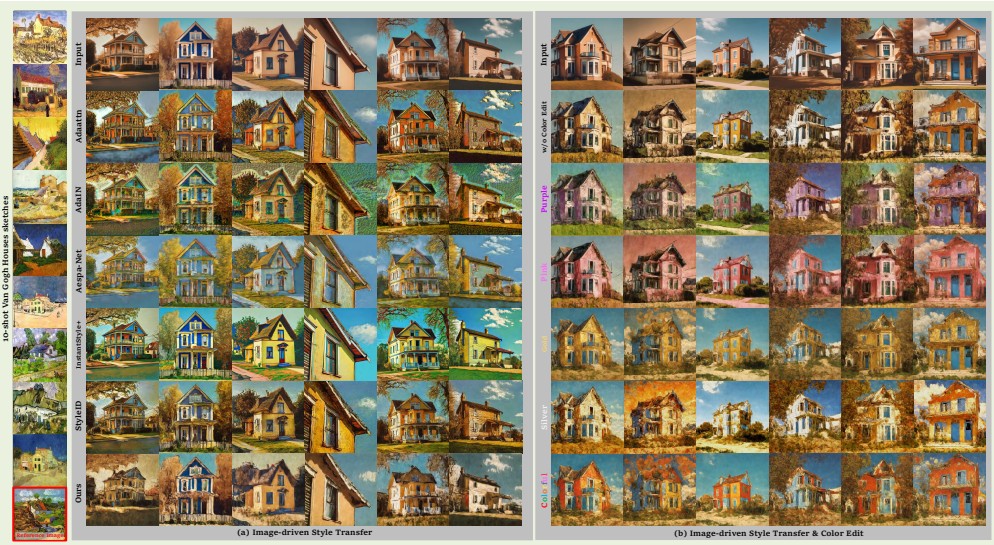

Figure 8: **Qualitative Comparison of Image-driven Style Transfer and Color Edit Results on Van Gogh houses Dataset Using Different Methods.** Due to page limitations, we have placed some of the experimental results in Appendix H.3 and H.4 and some comparison results with GPT-4o OpenAI (2024) in Appendix G.1.

| Method | House→Van Gogh style (text-driven stylization) | | | | | House→Monet style (text-driven stylization) | | | | | House→Cezanne style (text-driven stylization) | | | | |
|---|---|---|---|---|---|---|---|---|---|---|---|---|---|---|---|
| | CLIP-FID ↓ | DINO ↑ | CLIP-I ↑ | CLIP-T ↑ | LPIPS ↓ | CLIP-FID ↓ | DINO ↑ | CLIP-I ↑ | CLIP-T ↑ | LPIPS ↓ | CLIP-FID ↓ | DINO ↑ | CLIP-I ↑ | CLIP-T ↑ | LPIPS ↓ |
| Artist Jiang & Chen (2024) | 70.75 | 0.7925 | 0.6260 | 0.2989 | 0.8060 | 68.74 | 0.6699 | 0.4910 | 0.2755 | 0.7815 | 79.27 | 0.6587 | 0.5302 | 0.2830 | 0.7494 |
| InstructPix2Pix Brooks et al. (2023) | 72.36 | 0.7464 | 0.5680 | 0.2378 | 0.3677 | 85.05 | 0.6499 | 0.4696 | 0.2446 | 0.4135 | 77.23 | 0.6424 | 0.5336 | 0.2693 | 0.4151 |
| Plug-and-play Tumanyan et al. (2023) | 57.96 | 0.7977 | 0.6776 | 0.3086 | 0.4295 | 79.87 | 0.6638 | 0.5024 | 0.2545 | 0.2800 | 73.86 | 0.6576 | 0.5506 | 0.2777 | 0.3295 |
| Ours | 45.69 | 0.8075 | 0.6870 | 0.3117 | 0.7444 | 57.69 | 0.7049 | 0.5788 | 0.2859 | 0.7268 | 63.83 | 0.6761 | 0.5816 | 0.3145 | 0.7042 |

Table 2: **Quantitative Comparisons to SOTAs on Text-driven Stylization Task.**

## 3.4 ABLATION STUDY

To prove that the proposed techniques can indeed effectively improve the performance of *StyleWallfacer* in various generation scenarios, we conduct extensive ablation studies focusing on these techniques and leave them in Appendix E due to page limit. And we have also understood the source of *StyleWallfacer*'s superiority from a mathematical perspective, for details see Appendix D.

## 4 CONCLUSION

In this work, we focus on building a unified framework for style transfer by analyzing semantic drift, overfitting, and the true meaning of style transfer that previous works have failed to settle, and accordingly proposing a new method named *StyleWallfacer*. *StyleWallfacer* includes a one-stage fine-tuning process and a training-free inference framework that aims to solve these issues, namely the semantic-based style learning strategy, the training-free triple diffusion process, and the data augmentation method for small scale datasets based on human feedback. With these designs tailored to style transfer, our *StyleWallfacer* achieves convincing performance on text/image-driven style transfer scenarios, text-driven stylization, and image-driven style transfer with color edit, while solving problems before. In Appendix I and J, we will discuss possible limitations and potential future works of *StyleWallfacer*.

| Method | Mountain→Landscapes (one-shot) | | | | Houses→Van Gogh Houses (one-shot) | | | | Dogs→Watercolor Dogs (one-shot) | | | |
|---|---|---|---|---|---|---|---|---|---|---|---|---|
| | CLIP-FID ↓ | DINO ↑ | CLIP-I ↑ | LPIPS ↓ | CLIP-FID ↓ | DINO ↑ | CLIP-I ↑ | LPIPS ↓ | CLIP-FID ↓ | DINO ↑ | CLIP-I ↑ | LPIPS ↓ |
| AdaAttn Liu et al. (2021) | 60.92 | 0.7444 | 0.7200 | 0.7613 | 70.43 | 0.7839 | 0.5825 | 0.7046 | 40.05 | 0.7455 | 0.7556 | 0.6995 |
| AdaAIN Huang & Belongie (2017) | 64.03 | 0.7590 | 0.6942 | 0.7005 | 73.09 | 0.7892 | 0.5516 | 0.7504 | 37.82 | 0.7708 | 0.7530 | 0.7170 |
| AesPA-Net Hong et al. (2023) | 61.71 | 0.7554 | 0.6996 | 0.6592 | 65.65 | 0.7887 | 0.5979 | 0.7380 | 39.92 | 0.7438 | 0.7645 | 0.6677 |
| StyleID Chung et al. (2024) | 47.45 | 0.7518 | 0.7564 | 0.6062 | 55.79 | 0.7996 | 0.6501 | 0.7183 | 36.83 | 0.7615 | 0.7613 | 0.6859 |
| InstantStyle-Plus Wang et al. (2024c) | 59.04 | 0.7595 | 0.7381 | 0.3909 | 64.32 | 0.7582 | 0.6077 | 0.2903 | 41.04 | 0.7437 | 0.7633 | 0.3132 |
| Ours | 45.14 | 0.8124 | 0.8210 | 0.5917 | 37.19 | 0.8346 | 0.7309 | 0.7437 | 35.40 | 0.8041 | 0.7852 | 0.6848 |

Table 3: **Quantitative Comparisons to SOTAs on Image-driven Style Transfer Task.**

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

APPENDIX

OVERVIEW

This supplementary material provides the relsted works, additional experiments and results to further support our main findings and proposed *StyleWallfacer*. These were not included in the main paper due to the space limitations. The supplementary material is organized as follows:

# Contents

## A  RELATED WORKS

### A.1  ONE-SHOT TEXT-DRIVEN STYLE TRANSFER

The primary objective of one-shot text-driven image transfer is to generate images that are consistent with the content specified by a text prompt and the style of a single reference image. Recent advancements have concentrated on developing feature inversion techniques and implementing efficient fine-tuning strategies. In the context of feature inversion, Textual Inversion Gal et al. (2023) generates pseudo-words in the embedding space to represent specific styles. By embedding these pseudo-words during the generation process, the model can produce images that reflect the desired style. In contrast, data-driven strategies such as DreamBooth Ruiz et al. (2023) and SVDiff Han et al. (2023) bind image style information to a specific identifier through fine-tuning. During generation, the inclusion of this identifier in the text prompt enables the model to recognize and reproduce the associated style. However, when dealing with single-sample data, these methods are prone to issues such as model overfitting and semantic drift, which significantly impair the quality and diversity of the generated images. To address these challenges, this paper proposes a semantic-based style knowledge injection method and a human feedback-based small dataset enhancement strategy, effectively resolving the aforementioned limitations of traditional models.

### A.2  ONE-SHOT IMAGE-DRIVEN STYLE TRANSFER

The primary objective of one-shot image-driven style transfer is to perform style transfer on a target image based on the stylistic information extracted from a single reference style image. The main ideas of traditional one-shot image-driven style transfer methods lie in two aspects. The first is to use neural networks to extract high-level features from the style and content images and then fuse these features using a model. The second is to structure and encode the structural and stylistic information of the image and recombine the stylistic information of the style image with the structural information of the content image. A representative method for high-level feature fusion is AdaIN Huang & Belongie (2017), which aligns the mean and variance of the content features extracted by the neural network with those of the style features, thereby achieving instance-based style transfer results. However, the generated results are relatively coarse, with large areas of color blocks appearing in the images. To further refine the generated results, AesPA-Net Hong et al. (2023) uses a more advanced Transformer network to fuse the content and style features and employs an image refinement branch to optimize the generated results, making the stylized generation results more delicate. In terms of feature disentanglement and recombination, InstantStyle-Plus Wang et al. (2024c) uses ControlNet Zhang et al. (2023b) and DDIM Inversion to disentangle the structural information of the image and a style guide to extract information from the style image. Finally, based on these two parts of information, a pre-trained text-to-image model is used to generate the stylized image. StyleID Chung et al. (2024) uses DDIM Inversion to obtain the latent representations corresponding to the content and style images and fuses them using AdaIN to get the initial latent of the model. During the denoising process, the key and value in the self-attention layer are replaced with the key and value from the style latent denoising, and the original query is combined with the content latent to ultimately achieve high-quality stylized generation results. However, these methods all share a common drawback: they can only rigidly learn the texture and color features of the style image and simply replace the texture and color of the content image based on these features, resulting in generated images that lack true stylistic features and merely possess some of the textures and colors of the style image. The method proposed in this paper aims to address this issue and achieve artist-level style transfer results.

### A.3  TEXT-DRIVEN IMAGE STYLIZATION

The primary objective of text-driven image stylization is to perform style transfer on the target image based on the stylistic descriptions provided in the text prompt. This process primarily leverages the rich stylistic prior knowledge inherent in the pre-trained models. Prior to the widespread adoption of diffusion models, text-driven image manipulation was predominantly achieved by optimizing an image representation Kwon & Ye (2022); Michel et al. (2022); Wang et al. (2024a); Cai et al. (2023) or the distribution of images Gal et al. (2022); Kim et al. (2022); Cai et al. (2025), utilizing specific forms of CLIP loss Radford et al. (2021). Subsequently, it was demonstrated that text-to-image (T2I) diffusion models could be adapted for analogous optimization schemes Hertz et al. (2023); Jiang et al. (2023); Kawar et al. (2023); Poole et al. (2023); Cai et al. (2024). For instance, Instruct-Pix2Pix

Brooks et al. (2023) fine-tunes the diffusion model with a language model to facilitate generalized editing tasks. Diffstyler Huang et al. (2025) learns a content and style-specific denoiser to achieve disentanglement. FreeStyle Liu et al. (2024) modulates the U-Net features to enable training-free stylization. More recent research has shifted focus to stylized image generation, where the content is provided as a prompt Chen et al. (2023); Gao et al. (2024); Hertz et al. (2024); Wang et al. (2024b;c); Tian et al. (2023). However, these methods are not directly related to text-driven stylization. Recent work directly related to text-driven image stylization is Artist Jiang & Chen (2024), which employs DDIM inversion and attention-related operations to achieve style guidance and structure preservation during the model's generation process. However, these methods suffer from the problem of text-style mismatch, which is partly due to the failure of text guidance and partly due to the imbalance between content preservation and style transfer. This paper addresses the aforementioned issues through an innovative triple diffusion process, achieving high-quality text-driven image stylization.

## B  IMPLEMENTATION DETAIL

### B.1  MODEL

Our *StyleWallfacer* adopts Stable Diffusion as its foundational model. To ensure fairness, *StyleWallfacer* and all baseline methods utilize the same base model, specifically SDXL Podell et al. (2024). However, *StyleWallfacer* is likely compatible with newer versions of SD, as the fine-tuning techniques proposed in this work are not contingent upon the specific architecture of the current version. During the two-stage fine-tuning process, we apply LoRA from the PEFT Xu et al. (2023) framework to the UNet of SD, with a rank of $r = 8$. By default, LoRA is applied to the parameters $to\_k$, $to\_q$, $to\_v$, $to\_out.0$, $add\_k\_proj$, and $add\_v\_proj$. The text encoder $\tau$ is neither fine-tuned nor subjected to LoRA adaptation. When removing style-related descriptions from text using LLMs, we selected the Meta Llama-3.2-1B AI (2024). In reality, LLMs with larger parameter sizes might perform better in this regard. However, this goes beyond the scope of this study.

### B.2  TRAINING

For the first stage, we train the model for 1500 steps, with batch size 4 and learning rate $1 \times 10^{-4}$. During this periods, gradient checkpointing and 8bit Adam are also applied to save VRAM. All the experiments running *StyleWallfacer* in this work are done on four NVIDIA A100 GPUs with 80GB VRAM.

### B.3  INFERENCE

The proposed method in this paper is capable of performing a diverse range of image style transfer tasks. However, slight variations in the pipeline may exist between different tasks. This section will elaborate on the pipelines employed by *StyleWallfacer* for accomplishing distinct tasks. As shown in Figure 9, the pipelines employed by *StyleWallfacer* for different tasks are illustrated.

**Text-driven Style Transfer.** As shown in Figure 9 (a), when performing text-driven style transfer, the model employs the classic Stable Diffusion pipeline for generation, leveraging the style knowledge from the style LoRA to guide the model in producing images that match the text description. In this process, the text description does not need to include any style-related descriptors, such as "Van Gogh style". Instead, it should focus solely on describing the content and structure of the desired image.

**Text-driven Styliztion.** As shown in Figure 9 (b), when performing text-driven stylization, the model employs the triple diffusion pipeline designed in this paper for generation. There is no need to load LoRA weights into the U-Net. Simply describe the main content of the image to be transferred and add style-related trigger words. For example, if the content of the image to be transferred is a house and the user wishes to transfer it to the Van Gogh style, the prompt "A house in Van Gogh style" can be used, and the model will complete the style transfer task.

**Image-driven Style Transfer.** As shown in Figure 9 (b), When performing image-driven style transfer, the model also employs the triple diffusion pipeline for generation. But it is necessary to load the pre-trained LoRA weights corresponding to the desired style, as proposed in this paper, into the U-Net. During style transfer, a brief description of the image's main content should be provided

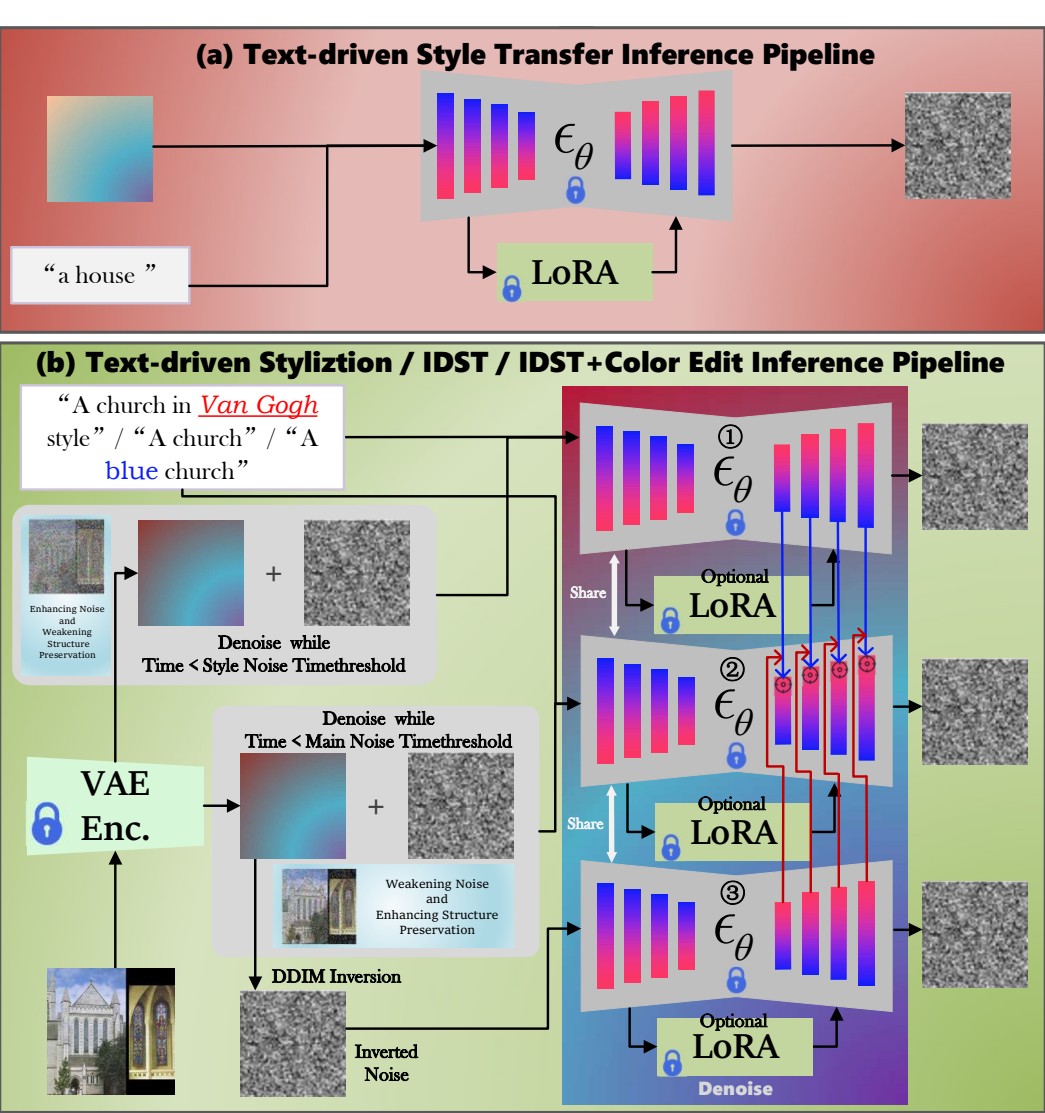

Figure 9: **Illustration of the Pipelines Employed by *StyleWallfacer* for Different Tasks.**

as the text prompt. There is no need to include any style-related trigger words; simply describe the basic content of the image to be transferred.

**Text-driven Style Transfer & Color Edit.** As shown in Figure 9 (b), for text-driven style transfer & color edit task, the same pipeline is used as for image-driven style transfer. Simply add color descriptions to the text prompt, such as "A blue house."

### B.4 DETAILS ABOUT THE BENCHMARK

Traditional one-shot style-transfer algorithms merely fuse the texture of a reference image onto a content image; consequently, the widely used one-shot benchmarks, despite their large style counts, only assess how well a model mimics that one reference.

In contrast, we treat an artist's style as an abstract concept emerging from commonalities across multiple works. Therefore, we deliberately discard the conventional evaluation mindset and instead adopt the few-shot protocols established in the literature. These protocols are more demanding, as they force the model to distill domain-level characteristics from just a handful of exemplars. Through this evaluation strategy, we can better assess the model's ability to learn abstract stylistic knowledge rather than merely focusing on superficial texture blending.

## C THE USE OF LARGE LANGUAGE MODELS

In this paper, large language models (LLMs) are primarily employed for caption generation and prompt refinement.

In LLM, we only use a single one-time user dialogue to complete the editing of the text description. Here are the relevant details about the dialogue:

*Sys*: You are now a master of style related text removal, and the task given to you is to remove the description of style from a text describing a picture. Directly complete the task I gave you, give the results and don't have repetitive answers.

*Sys*: Complete the task I gave you and do not give the process.

*User*: Remove the style related information from the description and output it in the same structure: *text*.

## D THE MATHEMATICAL EXPLANATION OF THE EFFICACY OF *StyleWallfacer*

To better explain the origin of *StyleWallfacer*'s superiority over other solutions, this section delves into a mathematical analysis of *StyleWallfacer*'s performance.

Mathematically, style transfer and image - editing tasks are essentially about transforming one data distribution $p_t(x)$ into another $q_t(y)$. Assuming each sample from the $p_t(x)$ distribution can be transformed into the $q_t(y)$ distribution, the above problem can be regarded as a continuous optimal transport problem. Therefore, we can utilize the theory of optimal transport to better understand *StyleWallfacer*. However, in the theory of optimal transport, the Wasserstein distance is mostly used to describe the distance between distributions, which does not have a direct connection with diffusion models. Consequently, we need to establish a relationship between the model and the W-distance to better assist us in theoretically investigating the mechanism of the model.

Suppose $p_t(x)$ follows the forward SDE process:

$$dx = f(x,t)dt + g(t)dw, \quad t \in [0,T] \tag{5}$$

Starting from $t = 0$, define $p_0(x)$ as the data distribution. Let $s_\theta(t,x)$ be trained via score matching loss from (5). Assume $q_t(x)$ follows the reverse SDE process:

$$dx = [f(x,t) - g(t)^2 s_\theta(x,t)]dt + g(t)dw, \quad t \in [0,T] \tag{6}$$

According to Theorem 8.4.7 of Ambrosio et al. (2008) and Proposition 5.25 of Santambrogio (2015), we have:

$$-\frac{1}{2}\frac{dW_2^2(p_t(x), q_t(y))}{dt} = \mathbb{E}_{\pi_t(x,y)}\left[(x - y) \cdot \left(\frac{dy}{dt} - \frac{dx}{dt}\right)\right] \tag{7}$$

where $\pi_t(x, y)$ denotes the optimal transport plan from $p_t(x)$ to $q_t(y)$, and $\frac{dx}{dt}$ and $\frac{dy}{dt}$ are the total derivatives of the paths $x$ and $y$ with respect to $t$, corresponding to the probability flow ODEs.

The probability flow ODE corresponding to (5) is:

$$\frac{dx}{dt} = f(x, t) - g(t)^2 \nabla_x \log p_t(x) \tag{8}$$

The probability flow ODE corresponding to (6) is:

$$\frac{dy}{dt} = f(y, t) - g(t)^2 s_\theta(y, t) + \frac{1}{2}g(t)^2 \nabla_y \log q_t(y) \tag{9}$$

Substituting these into (7), we get:

$$
\begin{aligned}
-\frac{1}{2}\frac{dW_2^2(p_t(x), q_t(y))}{dt} = {} & \mathbb{E}_{\pi_t(x,y)}[(x - y) \cdot (f(y, t) - f(x, t))] \\
& + g(t)^2 \mathbb{E}_{\pi_t(x,y)}\left[(x - y) \cdot (s_\theta(x, t) - s_\theta(y, t))\right] \\
& + g(t)^2 \mathbb{E}_{\pi_t(x,y)}\left[(x - y) \cdot (\log \nabla_x p_t(x) - s_\theta(x, t))\right] \\
& + \frac{g(t)^2}{2}\mathbb{E}_{\pi_t(x,y)}\left[(x - y) \cdot (\log \nabla_y q_t(y) - \log \nabla_x p_t(x))\right]
\end{aligned}
\tag{10}
$$

The first and second terms on the right-hand side can be easily obtained as:

$$
\begin{aligned}
\mathbb{E}_{\pi_t(x,y)}\left[(x - y) \cdot (f(y, t) - f(x, t))\right] &\leq L_f(t)\mathbb{E}_{\pi_t(x,y)}\left[\|x - y\|^2\right] \\
&= L_f(t)W_2^2(p_t(x), q_t(y))
\end{aligned}
\tag{11}
$$

and

$$
\begin{aligned}
g(t)^2 \mathbb{E}_{\pi_t(x,y)}\left[(x - y) \cdot (s_\theta(x, t) - s_\theta(y, t))\right] &\leq g(t)^2 L_s(t)\mathbb{E}_{\pi_t(x,y)}\left[\|x - y\|^2\right] \\
&= g(t)^2 L_s(t)W_2^2(p_t(x), q_t(y))
\end{aligned}
\tag{12}
$$

For the third term, using the integral Cauchy-Schwarz inequality:

$$
\begin{aligned}
& g(t)^2 \mathbb{E}_{\pi_t(x,y)}[(x - y) \cdot (\log \nabla_x p_t(x) - s_\theta(x, t))] \\
& \leq g(t)^2 \mathbb{E}_{\pi_t(x,y)}[\|x - y\|]^{\frac{1}{2}}\mathbb{E}_{\pi_t(x,y)}[\|\log \nabla_x p_t(x) - s_\theta(x, t)\|^2]^{\frac{1}{2}} \\
& = g(t)^2 W_2(p_t(x), q_t(y))\mathbb{E}_{p_t(x)}[\|\log \nabla_x p_t(x) - s_\theta(x, t)\|^2]^{\frac{1}{2}}
\end{aligned}
\tag{13}
$$

For the fourth term:

$$\mathbb{E}_{\pi_t(x,y)}[(x - y) \cdot (\log \nabla_y q_t(y) - \log \nabla_x p_t(x))] \leq 0 \tag{14}$$

Combining (11) to (14), we finally get:

$$
\begin{aligned}
-\frac{1}{2}\frac{dW_2^2(p_t(x), q_t(y))}{dt} \leq {} & L_f(t)W_2^2(p_t(x), q_t(y)) \\
& + g(t)^2 L_s(t)W_2^2(p_t(x), q_t(y)) \\
& + g(t)^2 W_2(p_t(x), q_t(y))b_t^{\frac{1}{2}}
\end{aligned}
\tag{15}
$$

where $b_t = \mathbb{E}_{p_t(x)}[\|\log \nabla_x p_t(x) - s_\theta(x, t)\|^2]$.

Due to:

$$-\frac{1}{2}\frac{dW_2^2(p_t(x), q_t(y))}{dt} = -W_2(p_t(x), q_t(y))\frac{dW_2(p_t(x), q_t(y))}{dt} \tag{16}$$

Rearranging both sides of (15):

$$-\frac{dW_2(p_t(x), q_t(y))}{dt} \le (L_f(t) + g(t)^2 L_s(t))W_2(p_t(x), q_t(y)) + g(t)^2 b_t^{\frac{1}{2}} \tag{17}$$

This is a first-order non-linear differential inequality. Using the method of integrating factors, let:

$$W_2(p_t(x), q_t(y)) = C_t \exp\left(\int_t^0 L_f(r) + g(r)^2 L_s(r)dr\right) = C_t/I(t) \tag{18}$$

Substituting into (16):

$$-\frac{dC_t}{dt} \le \exp\left(\int_0^t L_f(r) + g(r)^2 L_s(r)dr\right)g(t)^2 b_t^{\frac{1}{2}} = I(t)g(t)^2 b_t^{\frac{1}{2}} \tag{19}$$

Integrating both sides from $t = 0$ to $T$:

$$C_0 \le \int_0^T I(t)g(t)^2 b_t^{\frac{1}{2}}dt + C_T \tag{20}$$

Since $W_2(p_T(x), q_T(y)) = C_T/I(T)$, we have $C_T = I(T)W_2(p_T(x), q_T(y))$. Therefore:

$$W_2(p_0(x), q_0(y)) = C_0 \le \int_0^T I(t)g(t)^2 b_t^{\frac{1}{2}}dt + I(T)W_2(p_T(x), q_T(y)) \tag{21}$$

**It indicates that optimizing the score matching loss is equivalent to optimizing the Wasserstein distance.** But diffusion models often struggle to follow the shortest Wasserstein distance path during distribution transformation, resorting to a near-optimal route instead. For different models, this path can deviate from the optimal in both direction and distance, causing flawed distribution transformation and subpar generation results.

Thus, there are significant relationships between the efficiency and quality of distribution transformation in diffusion models during style transfer and image editing, and the Wasserstein distance. But the relationships between image distributions are complex and hard to study directly. In the case of a single - point distribution in two - dimensional space, the Wasserstein distance equals the corresponding Euclidean distance. So, to better understand this theory, we use t-SNE to reduce the dimension of *StyleWallfacer*'s image-driven style transfer results on the Van Gogh houses dataset, along with results from other methods, to two-dimensional. Then, we use visualizations for intuitive analysis.

As shown in Figure 10, this is a schematic diagram of the initial image before style transfer and ten reference style images after t-SNE dimensional reduction. In the two - dimensional distribution transformation, the straight lines are the optimal routes for distribution transformation (shown as red lines in the figure). Therefore, the generative model should ensure that the generation process follows the optimal transport path as much as possible to achieve the best results.

As shown in Figure 11, the distribution transformation process in two - dimensional space is often represented by a curve. When the parameters of *StyleWallfacer* are not optimal, the model's distribution transformation deviates in both direction and distance from the target distribution (e.g., red for $t_s^s = 200$ and purple for $t_s^s = 900$). However, when parameters are optimized (green), the process aligns with the optimal transport path, ensuring generated results match the target distribution.

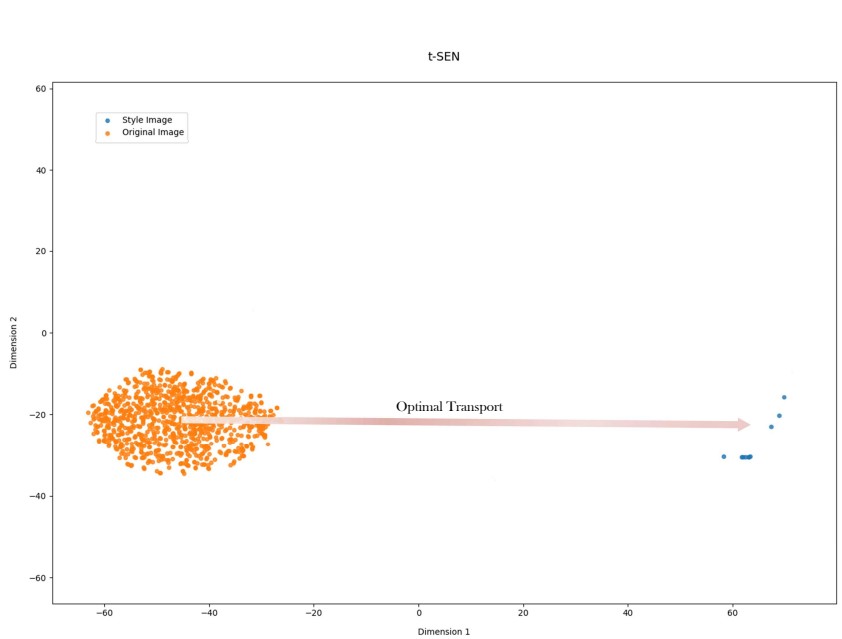

Figure 10: **The Performance of *StyleWallfacer* in Image Editing Tasks.**

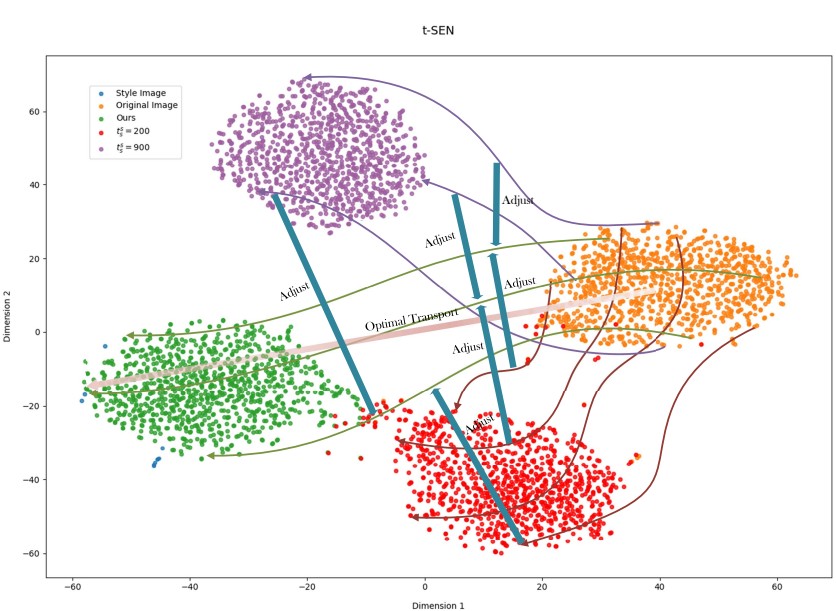

Figure 11: **The Performance of *StyleWallfacer* in Image Editing Tasks.**

This highlights *StyleWallfacer*'s unique ability to ensure accurate distribution transformation through hyperparameter tuning, setting it apart from other models.

As shown in Figure 12, traditional style transfer methods deviate significantly from the optimal transport path. This deviation occurs in both distance and direction. Even better-performing methods like InstantStyle-Plus and StyleID, while close to the optimal path, still show slight deviations. In contrast, *StyleWallfacer* accurately follows the optimal transport path, transferring the original distribution to the target one precisely. This highlights *StyleWallfacer*'s advantage: its precise distribution transfer control. By adjusting hyperparameters, it ensures accurate distribution transfer and avoids potential deviations.

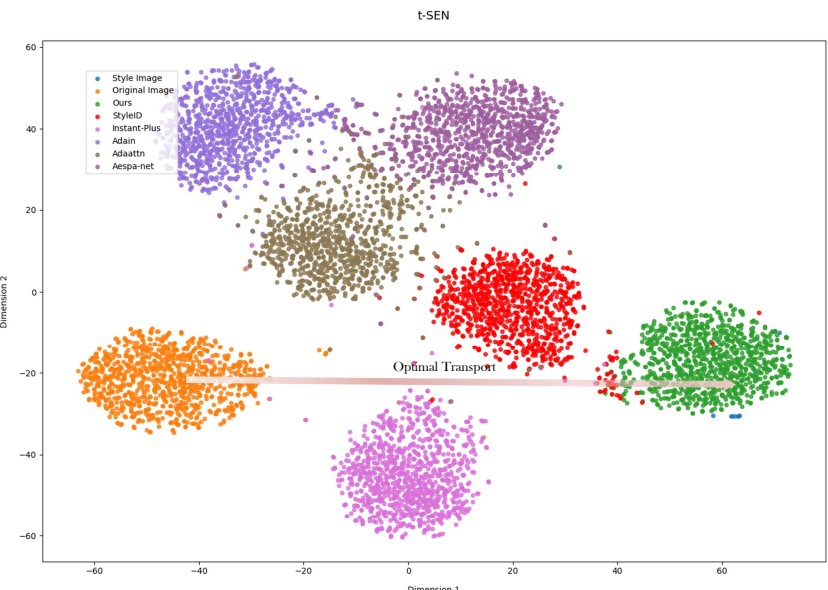

Figure 12: **The Performance of *StyleWallfacer* in Image Editing Tasks.**

# E    ABLATION STUDY

## E.1    STUDY ON THE ROLE OF *Key* AND *Value* IN STYLE AND SEMANTIC GUIDANCE

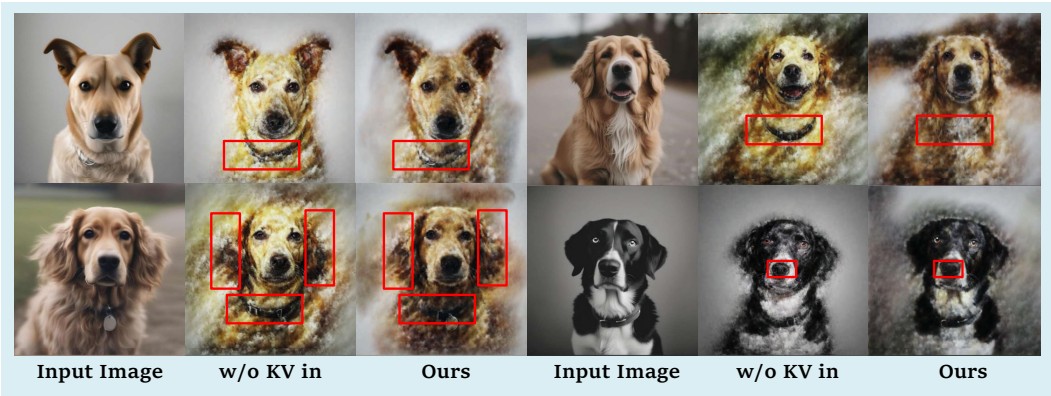

Figure 13: **Schematic comparison of style transfer results without *Key-Value* replacement versus *StyleWallfacer*.**

| Method | Dogs→Watercolor Dogs (one-shot) | | | |
| --- | --- | --- | --- | --- |
| | CLIP-FID ↓ | DINO ↑ | CLIP-I ↑ | LPIPS ↓ |
| w/o KV In | 35.61 | 0.7945 | 0.7926 | 0.7913 |
| Ours | 35.40 | 0.8041 | 0.7852 | 0.6848 |

Table 4: **Quantitative Comparisons to w/o KV In Version on Image-driven Style Transfer Task on Watercolor Dogs Dataset.**

In fact, as illustrated in Figure 13, the model can still accomplish the style transfer task even without the guidance of *Key* and *Value*. Moreover, when not compared with our proposed method, its style transfer results significantly outperform traditional approaches. However, the critical flaw of abandoning *Key-Value* guidance lies in the model's inability to achieve an optimal balance between content preservation and stylization—unlike the full implementation of *StyleWallfacer*. Regardless of parameter adjustments, methods without *Key-Value* guidance consistently suffer from content mismatch in the stylized output relative to the original image. For instance, in the figure, the shape of the dog's chain is altered, and the breed of the dog changes. In contrast, *StyleWallfacer*, enhanced by triple diffusion guidance, employs a relatively small t in the primary diffusion process to ensure robust content preservation. As shown in Table 4, when the model without the guidance of *Key* and *Value*, it experiences a significant drop in the LPIPS metric, which indicates a substantial compromise in its ability to preserve image content. In terms of style similarity, it only outperforms *StyleWallfacer* in the CLIP-I score, proving that this mode fails to balance content preservation and style transfer. Meanwhile, in the complete *StyleWallfacer* architecture, *Key-Value* guidance from diffusion processes with larger $t_s$ further strengthens the model's stylization performance, thereby achieving the best trade-off between content fidelity and stylistic expression.

## E.2 STUDY ON THE ROLE OF *Query* IN IMAGE CONTENT PRESERVATION

To investigate the role of *Query* in achieving image content preservation during the generation process, this paper designs experiments by adjusting the hyperparameter $\beta$, and obtains the corresponding research results through visual comparisons. When the value of the hyperparameter $\beta$ is larger, it indicates that the *Query* extracted by DDIM inversion has a greater proportion in the fused *Query*. Conversely, when the value of $\beta$ is smaller, it means that the original *Query* has a larger proportion.

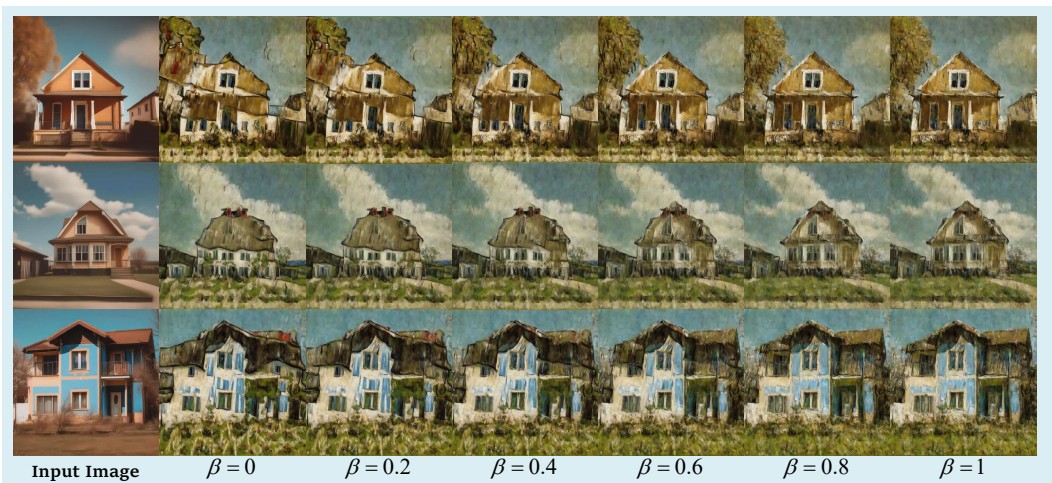

Figure 14: **Visualization of the Impact of Varying $\beta$ on Model Generation When $t_s^l$ and $t_s^s$ Are Fixed.**

As shown in Figure 14 and 15, when the values of $t_s^l$ and $t_s^s$ are fixed at 800 and 600 respectively (which is not the optimal state), a larger value of $\beta$ results in the generated image having a structure and content that are more similar to the original image. Conversely, when the value of $\beta$ is smaller, the preservation of the image structure deteriorates. Therefore, by adjusting the proportion of the original

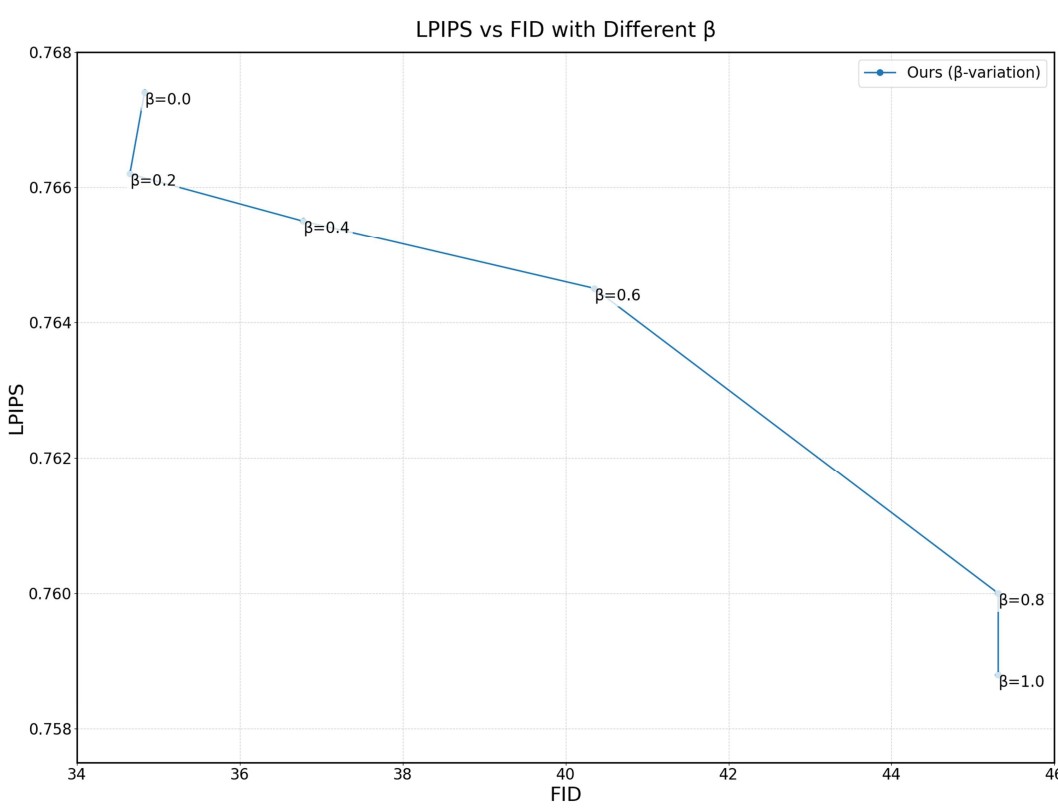

Figure 15: **Visualization of the Impact of Varying $\beta$ on LPIPS and CLIP-FID When $t_s^l$ and $t_s^s$ Are Fixed.**

*Query* and the DDIM inverted *Query*, one can effectively control the similarity of the generated image's content to the original image. This further confirms the effectiveness of the proposed method of preserving image structure through *Query*, as introduced in this paper.

Secondly, this experiment indirectly corroborates that the reason why the latent obtained through DDIM Inversion can maintain the original image during the denoising process is largely due to the role of the *Query*. Specifically, during the denoising of the inverted latent, the *Query* enables the model to focus more effectively on the intrinsic information of the image. It is precisely this characteristic that allows the extracted *Query* to assist the primary diffusion process proposed in this paper in preserving the content and structure of the image.

### E.3 STUDY ON THE IMPACT OF NOISE TIME THRESHOLDS ON MODEL GENERATION OUTCOMES

As discussed in the main text, this paper regards noise time thresholds $t$ as the "regulator" for balancing style transfer and content preservation, playing a crucial role in *StyleWallfacer*. Therefore, to further investigate the role of Noise Time Thresholds in the style transfer process, this section conducts in-depth experiments to explore it.

#### E.3.1 STUDY ON THE $t_s^l$

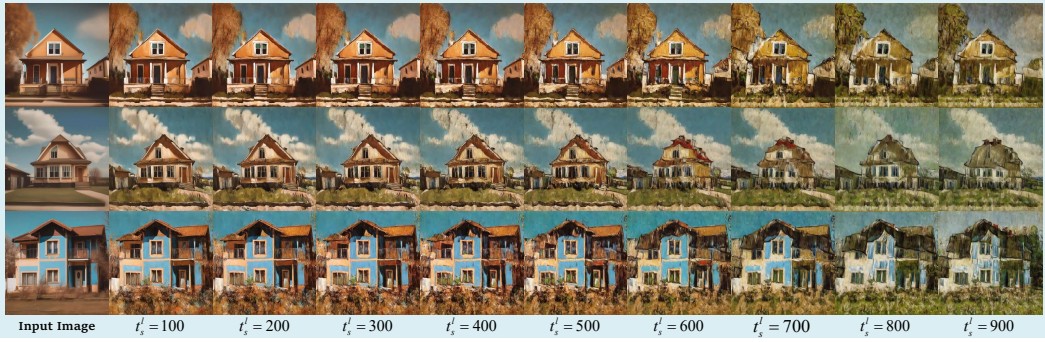

Figure 16: **Visualization of the Impact of Varying $t_s^l$ on Model Generation When $\beta$ and $t_s^s$ Are Fixed.**

As shown in Figure 16, when we fix the values of $t_s^s$ and $\beta$ at 600 and 0.5 respectively, the style transfer results gradually become more stylized while neglecting the content and color preservation of the original image as $t_s^l$ increases. Conversely, when $t_s^l$ is too small, the stylization results are not satisfactory. Unlike the role of $\beta$, which regulates the content preservation of the image but cannot achieve color preservation (as shown in Figure 14, no matter how $\beta$ is adjusted, it cannot make the stylized result similar to the original image in terms of color), it is necessary to adjust $t_s^s$ to modify the color space of the stylized image. Combining this with the regulation of *Query* by $\beta$ can achieve more realistic style transfer while better preserving the content and color.

As shown in Figure 17, when $t_s^s$ and beta are fixed, the overall trends of LPIPS and FID, as well as the visual effects, are consistent as $t_s^l$ varies. However, when $t_s^l$ exceeds 600, an inverse increase in FID is observed. This occurs because when the model becomes overly stylized, the color distribution of the generated results tends to become monotonous. As a result, when calculating KID, samples with similar colors exhibit high style similarity, while those with significantly different color distributions show low style similarity. Therefore, when adjusting the parameters, it is necessary to keep $t_s^l$ within a reasonable range, neither too large nor too small. An excessively large $t_s^l$ can lead to a decrease in the color diversity of the generated results, while an overly small $t_s^l$ can result in insufficient stylization of the model's output.

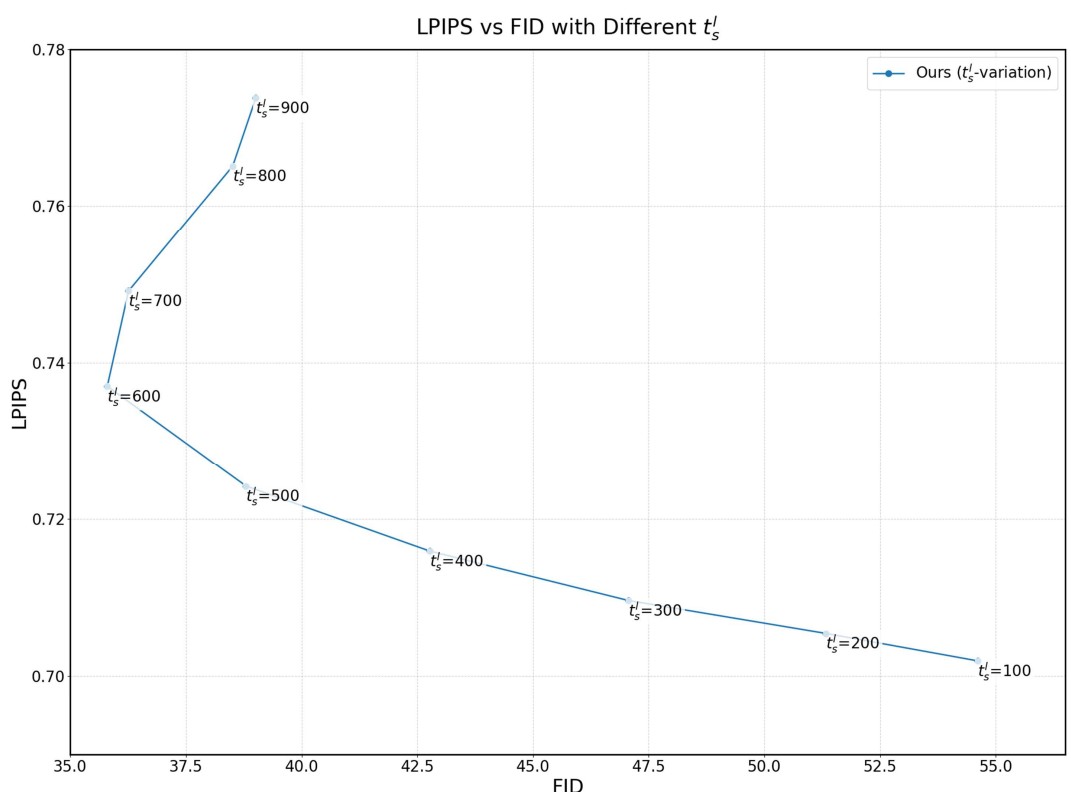

Figure 17: **Visualization of the Impact of Varying $t_s^l$ on LPIPS and CLIP-FID When $\beta$ and $t_s^s$ Are Fixed.**

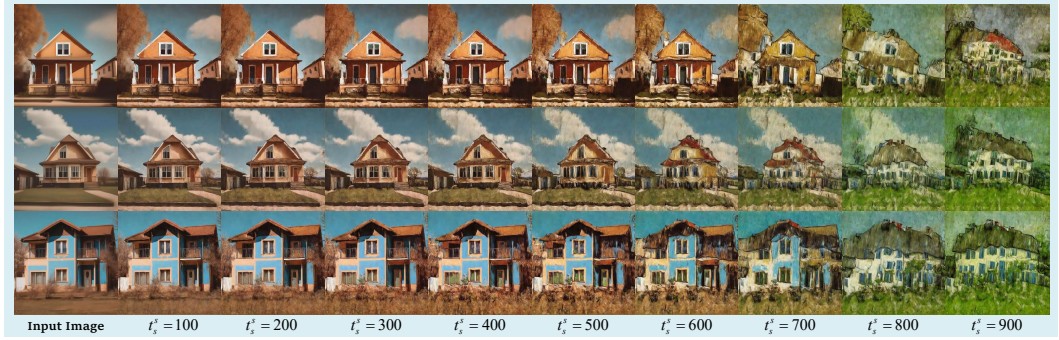

Figure 18: **Visualization of the Impact of Varying $t_s^s$ on Model Generation When $\beta$ and $t_s^l$ Are Fixed.**

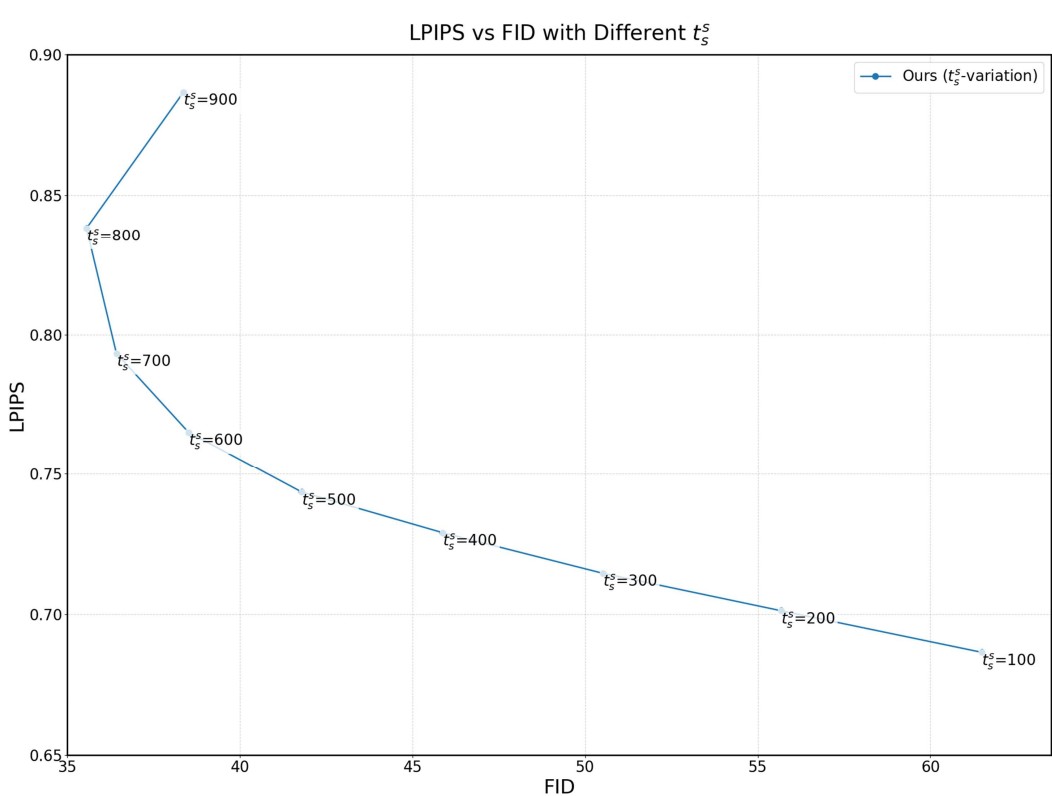

Figure 19: **Visualization of the Impact of Varying $t_s^s$ on LPIPS and CLIP-FID When $\beta$ and $t_s^l$ Are Fixed.**

### E.3.2 STUDY ON THE $t_s^s$

As shown in Figure 18, when we fix the values of $t_s^l$ and $\beta$ at 800 and 0.5 respectively, the style transfer results are gradually becoming more stylized while neglecting the content and color preservation of the original image as $t_s^s$ increases. However, unlike $t_s^l$, when $t_s^s$ gradually increases to a certain extent, the model's style transfer results will become completely inconsistent with the original image in terms of content. In contrast, when $t_s^l$ is larger, the model's style transfer results are simply more stylized. It can be seen that $t_s^s$ plays a crucial role in adjusting the consistency between the style transfer results and the original image content. When $t_s^s$ is too small, the model's generated results will tend more towards the original image. Conversely, when $t_s^s$ is too large, the model's generated results are overly stylized, leading to issues with content preservation. Therefore, a more moderate value of $t_s^s$ should be chosen to better cooperate with $t_s^l$ to achieve a balance between stylization and content preservation.

As shown in Figure 19, similar to the LPIPS changes dominated by $t_s^l$, the smaller the $t_s^s$, the more similar the model's stylized generation results will be to the content image. However, FID shows a significant increase when $t_s^s$ is greater than $t_s^l$. This is because an excessively large $t_s^s$ causes the content-guided process dominated by $t_s^s$ to converge with the stylization-guided process dominated by $t_s^l$, leading to the failure of content guidance. As a result, the model's stylized results become inconsistent in content and monotonous in stylized colors. Combining this with the previous analysis of $t_s^l$, when performing stylized generation, it is important to balance $t_s^l$ and $t_s^s$ to achieve a balance between content preservation and stylization in the generated results.

### E.4 GENERALIZABILITY STUDY ON IMAGE EDIT

Since *StyleWallfacer* has for the first time achieved color editing while completing image style transfer during image-driven style transfer, we speculate that the idea of *StyleWallfacer* can be applied to more image editing-related work.

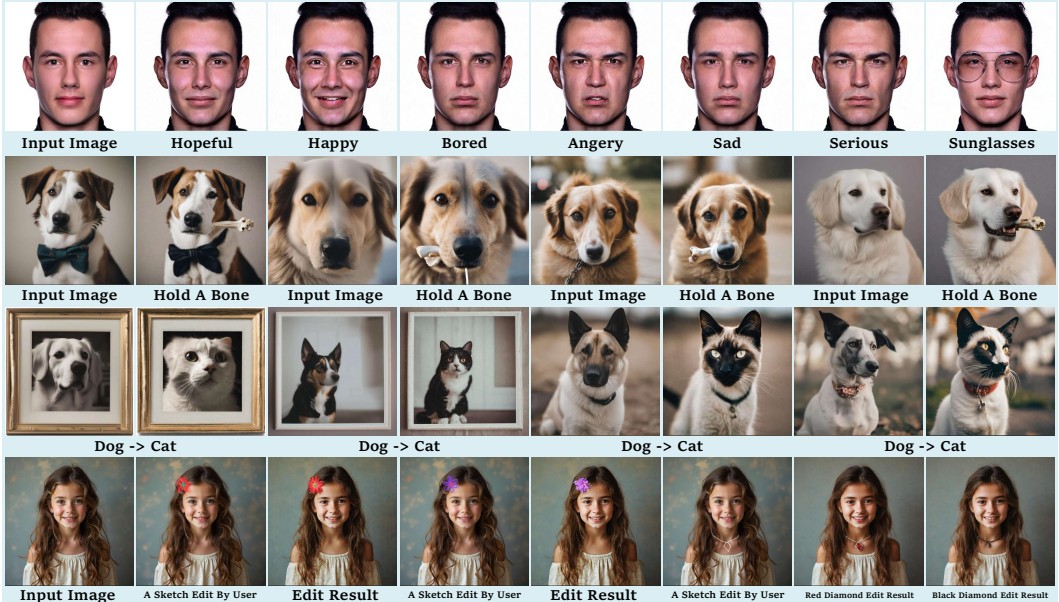

Figure 20: **The Performance of *StyleWallfacer* in Image Editing Tasks.**

As shown in Figure 20, *StyleWallfacer* also demonstrates good versatility in image editing tasks. By adjusting the values of $\beta$, $t_s^s$, and $t_s^l$, it can not only achieve conceptual image editing (such as the understanding and editing of expressions and facial features) but also produce object-driven editing results (such as sunglasses). Moreover, while accomplishing the aforementioned tasks, it can simultaneously maintain a good balance between image structure preservation and targeted editing.

Although *StyleWallfacer* still has room for improvement compared to some methods based on precise masks, such as more accurate content preservation and image editing to maintain complete consistency of the unedited parts with the original image, these minor differences are entirely acceptable given that *StyleWallfacerr* is training-free compared to those methods. Furthermore, future work can explore how to incorporate masks into the *StyleWallfacer* architecture to achieve more precise image editing and content preservation. This further proves the great application potential of the *StyleWallfacer* architecture.

## F PROBLEMS OF EXISTING METHODS AND THEIR VISUALIZATIONS

### F.1 LIMITED COLOR DOMAIN

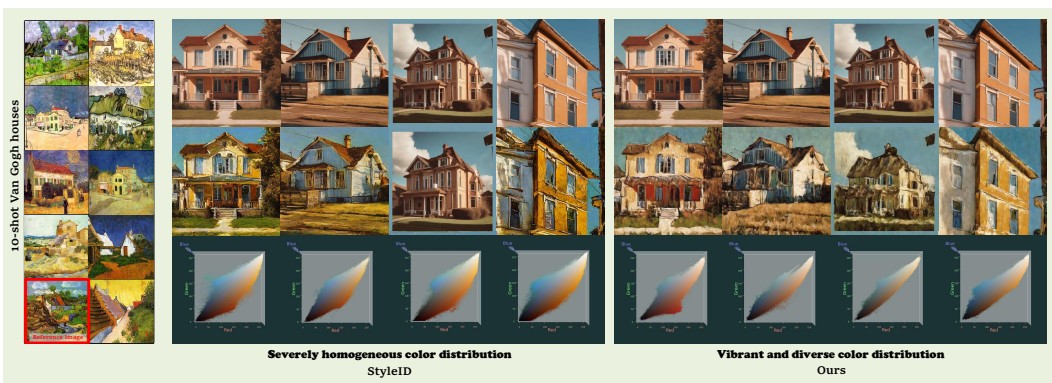

Figure 21: **Schematic Diagram of Limited Color Domain in Conventional Methods.**

As shown in Figure 21, most one-shot image-driven style transfer methods based on a single reference style image currently suffer from the problem that the color distribution of the generated image after style transfer is highly similar to that of the reference style image or that the color distributions of the generated images tend to converge. However, from the perspective of artistic creation, style transfer is closer to "imitating the creative style of a painter or artist" rather than mechanically combining the textures and colors of the reference style image with the target image to be transferred. Therefore, how to learn the true stylistic knowledge from a single reference style image and to "recreate" the target image based on style is the real problem that style transfer should address. In this way, style transfer is no longer about mechanically learning texture information, but about learning the true creative style of an artist, thereby generating digital works of real artistic value.

### F.2 FAILURE OF TEXT GUIDANCE

As shown in Figure 22, when using the prompt "*a sks church with no house around it, in a garden, pink sky*" for image generation, traditional methods generally produce images that do not match the text. This mismatch is not limited to a few samples, indicating that the problem has penetrated the model's semantic space and weakened its text control ability. This loss of control is manifested in aspects such as color inaccuracy, content deviation, and semantic drift.

In contrast, the model fine-tuned using the method proposed in this paper not only achieves consistency in style with the reference image but also realizes complete alignment between the generated results and the text prompt. Compared with other solutions, our method demonstrates superior text control, which enables the model to perform exceptionally well in tasks that require strict adherence to the text prompt, such as color editing.

### F.3 RISK OF OVERFITTING

As shown in Figure 23, the results presented are generated by training a conventional method using a single style image (the image within the red box) from the Van Gogh Houses dataset. Although these

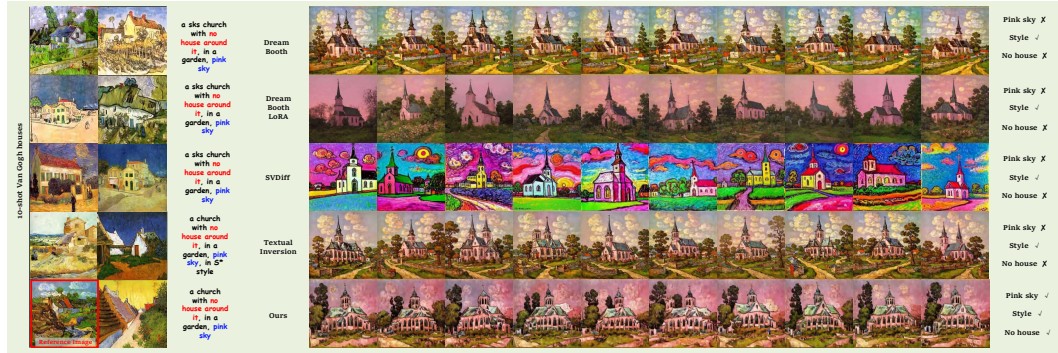

Figure 22: **Schematic Diagram of Text Control Failure in Conventional Methods.**

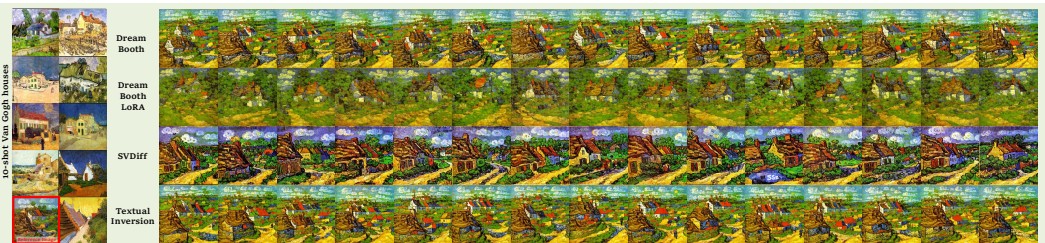

Figure 23: **Schematic Diagram of the Overfitting Problem in Conventional Methods.**

results vary in detail due to the different random seeds used during generation, their image structures, color distributions, and main styles are remarkably consistent. This is a manifestation of the model's overfitting.

Conventional methods, lacking effective data augmentation techniques, tend to drive the model towards two extremes during training: one is that the model does not overfit but fails to learn the true stylistic information; the other is that the model learns the stylistic information but suffers from severe overfitting. Traditional methods are unable to strike a good balance between these two extremes, that is, to enable the model to learn stylistic knowledge while retaining its diverse generation capabilities.

In this paper, we design a data augmentation strategy based on human feedback for small datasets. This approach effectively prevents the model from veering towards these extremes during training and balances the relationship between stylization and diversity.

# G  ADDITIONAL ANALYSIS

## G.1  COMPARISON WITH GPT-4O OPENAI (2024) IMAGE GENERATION

As one of the most popular models in the field of image generation, GPT-4o is hailed as a new solution for image generation and has demonstrated satisfactory results in multiple tasks. Therefore, to better explore the superiority of *StyleWallfacer*, this section conducts comparative experiments with GPT-4o.

As shown in Figure 24, the visualization results of GPT-4o and *StyleWallfacer* on the Van Gogh Houses dataset are presented. Clearly, *StyleWallfacer*'s style transfer results are more faithful to the reference style image's style compared to GPT-4o. For instance, in outlining house lines, *StyleWallfacer* uses black lines closer to the style image, whereas GPT-4o generates results with oil painting brushstrokes. GPT-4o seems to understand the reference image's style semantics as "Van Gogh style" and transfers it to the target image based on this semantic understanding and the model's prior knowledge of Van Gogh's style. This leads to GPT-4o's results being less faithful to the original

Figure 24: **Qualitative Comparisons to GPT-4o on Image-driven Style Transfer Task on Van Gogh Houses Dataset.**

style image and more similar to the texture and style characteristics of Van Gogh's more famous Starry Night painting.

| Method | Houses→Van Gogh Houses (one-shot) | | | |
| --- | --- | --- | --- | --- |
| | CLIP-FID ↓ | DINO ↑ | CLIP-I ↑ | LPIPS ↓ |
| GPT-4o OpenAI (2024) | 51.55 | 0.8019 | 0.6865 | 0.7647 |
| Ours | 41.26 | 0.8353 | 0.7056 | 0.7500 |

Table 5: **Quantitative Comparisons to GPT-4o on Image-driven Style Transfer Task on Van Gogh Houses Dataset.**

As shown in Table 5, the quantitative evaluation results of GPT-4o and *StyleWallfacer* on the Van Gogh Houses dataset are presented. Clearly, *StyleWallfacer* significantly outperforms GPT-4o in all four metrics, leading in both style similarity and content preservation.

# H  ADDITIONAL RESULTS

## H.1  ONE-SHOT TEXT-DRIVEN STYLE TRANSFER

As shown in Figure 25, *StyleWallfacer*'s generative results on three datasets are presented. On the Van Gogh houses dataset, our method achieves optimal style similarity and diversity, as discussed in the text. It is the only method that effectively transfers style knowledge across domains without confusion. In comparison, other methods suffer from semantic drift (DB, DB LoRA), incomplete style knowledge transfer (SVDiff), or style errors (Textual Inversion). This leads to various chaotic phenomena in the generated results and very poor quality of the produced images. For instance, the generated images may include elements of rifles, have inconsistent styles with the reference style, or even fail to complete the stylization process. Similar issues, along with severe overfitting, are observed in the watercolor dogs and landscape datasets. This demonstrates that our method efficiently addresses problems in text-driven style transfer using a single - style image, delivering excellent performance.

## H.2  TEXT-DRIVEN STYLIZTION

As shown in Figure 26, our *StyleWallfacer* proposed in this paper more faithfully reproduces the painter's creative style compared to other methods. For example, in the Cezanne style, only *StyleWallfacer* achieves the characteristics of Paul Cézanne's paintings, which are characterized by geometrical shaping and unique spatial structure. It emphasizes expressing the structure and spatial relationships of objects through the contrast and harmony of colors. Focusing on subjective expression, it pursues the purity and formal beauty of painting.

In the Monet style, *StyleWallfacer* provides a more realistic depiction of vegetation, gardens, and similar elements. This is particularly significant given Monet's painting style, which is quintessential Impressionism. Monet focused on capturing the fleeting effects of light and color through short, visible brushstrokes, often painting en plein air to directly observe and depict nature. His works

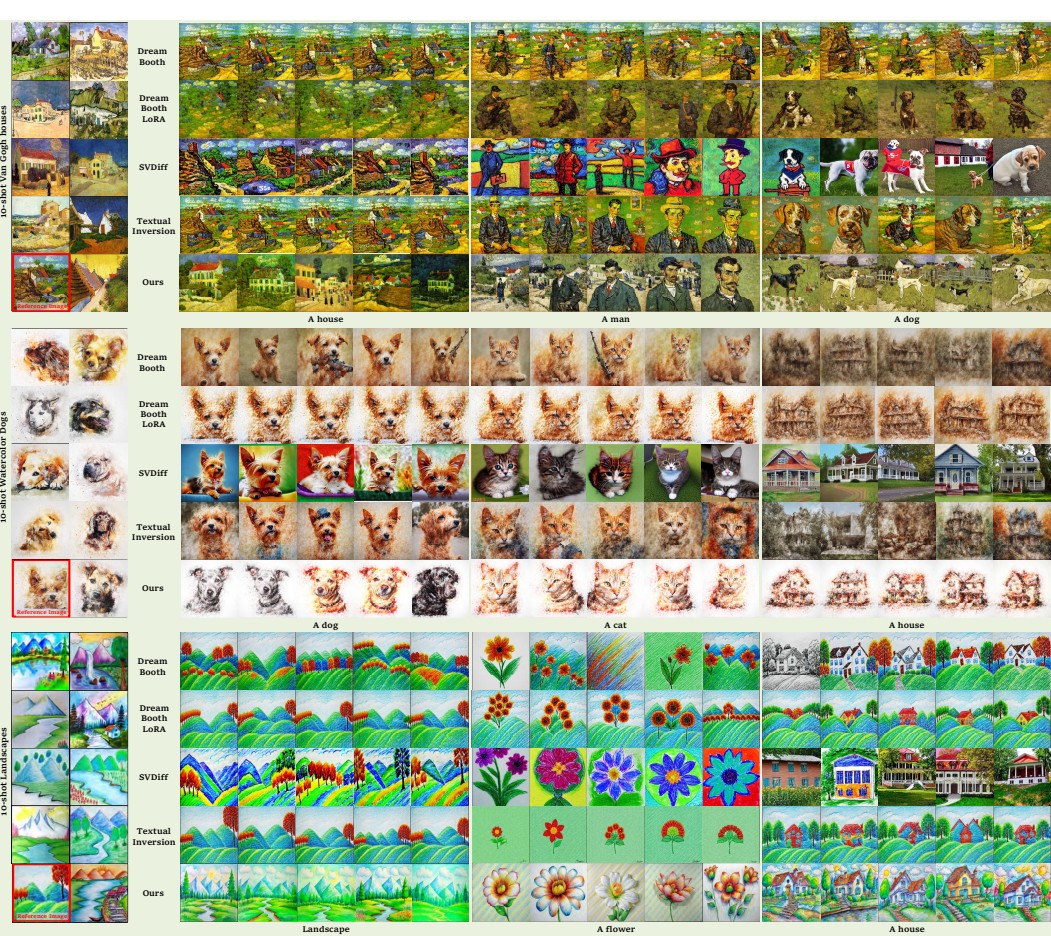

Figure 25: **Qualitative Comparison of Text-driven Style Transfer Results on Three Datasets Using Different Methods.**

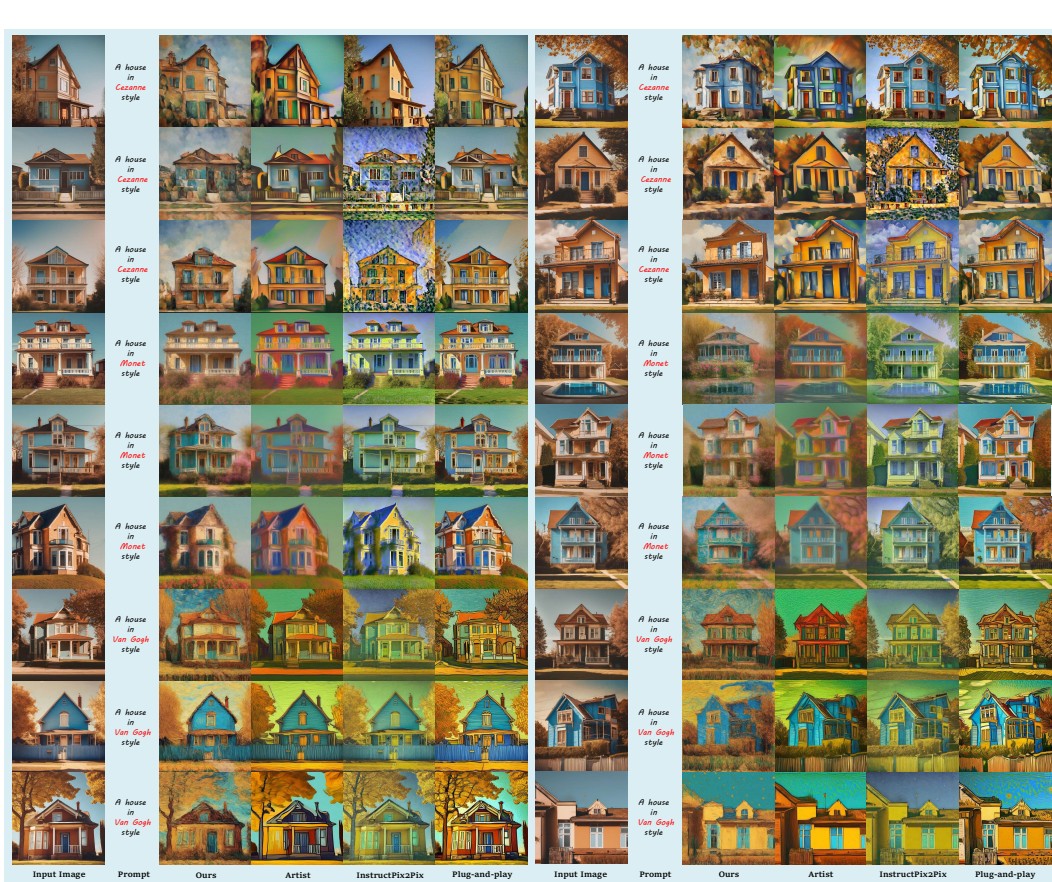

Figure 26: **Qualitative Comparison of Text-driven Stylization Results Using Different Methods.**

emphasize the interplay of light and atmosphere, with a particular interest in how light changes the appearance of objects. Similarly, *StyleWallfacer* excels in capturing these nuances, offering a more authentic representation of Monet's style compared to other methods.

In the Van Gogh style, *StyleWallfacer* excels in capturing the unique characteristics of his paintings. Van Gogh's art is defined by several key features: the use of "vivid and bold colors", often with striking contrasts; "thick and expressive brushstrokes" that convey a sense of raw emotion; and a "dynamic composition" that breaks away from traditional norms. *StyleWallfacer* captures these elements more authentically than other methods, providing a more vivid and emotionally resonant representation of Van Gogh's style.

## H.3 ONE-SHOT IMAGE-DRIVEN STYLE TRANSFER

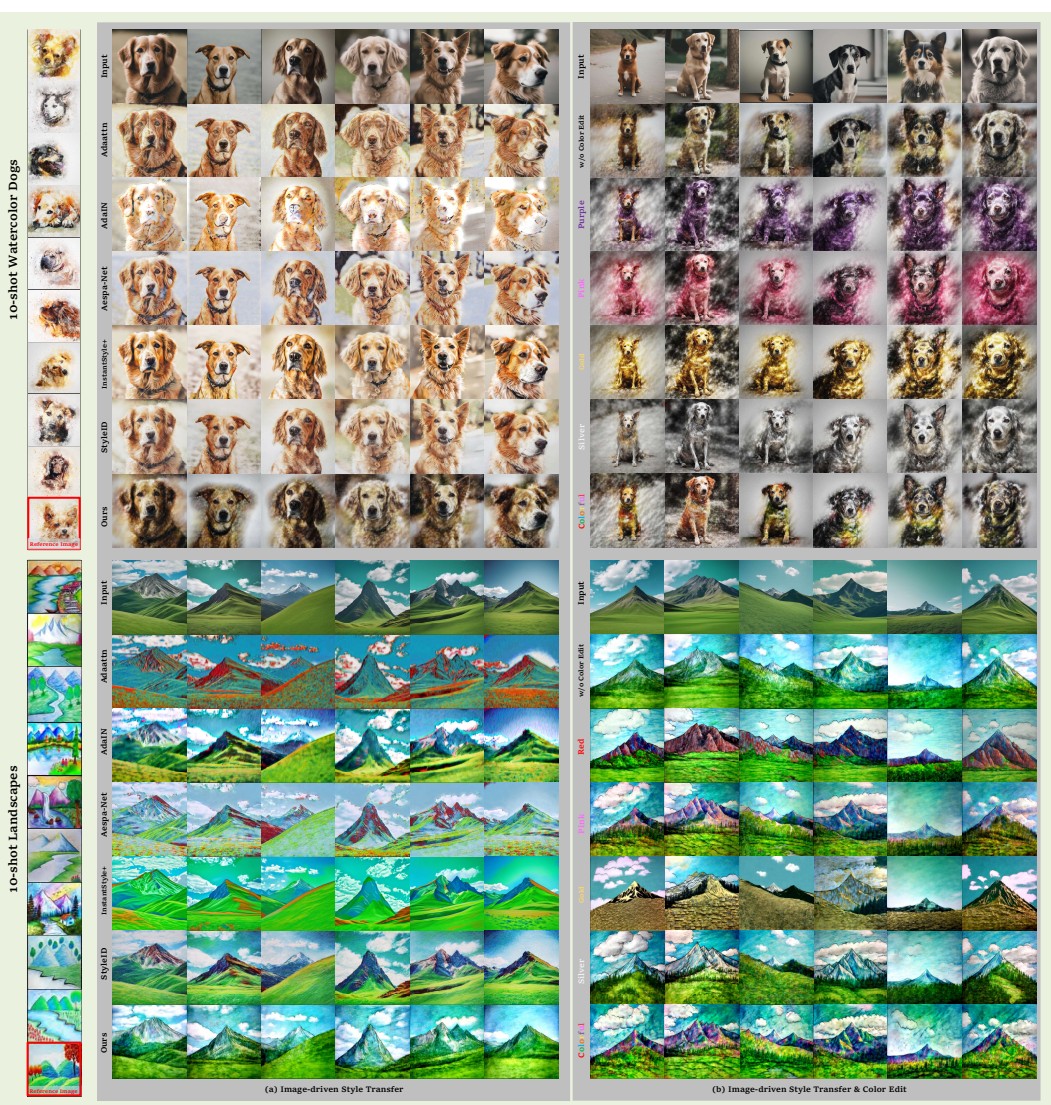

Figure 27: **Qualitative Comparison of Image-driven Style Transfer and Color Edit Results on Watercolor Dogs Datasets and Landscapes Datasets Using Different Methods.**

As shown in Figure 27, these are the test results of *StyleWallfacer*'s image-driven style transfer and color edit on the Watercolor Dogs dataset and the Landscapes dataset. Clearly, on the Watercolor Dogs dataset, due to the reference style images having stylistic features beyond texture (for example, a halo of watercolor bleeding around the puppies), traditional methods struggle to learn such abstract stylistic

features. This results in generated images that, while similar in color to the reference style images, only perform simple texture fusion in other style-related aspects (such as the choice of watercolor bleeding style and background color). Some even lose the unique brushstroke characteristics of watercolor, resembling oil painting instead. In contrast, *StyleWallfacer* retains a diverse color representation while fully learning the artistic expression of the reference style. It achieves the best style transfer results by considering both the watercolor brushstrokes and the compositional features that highlight the subject and downplay the background.

On the Landscape dataset, the overall artistic style is more abstract, which leads to traditional methods failing to understand the stylistic features of the reference images during style learning (resulting in simple color or texture fusion). This issue is particularly evident in methods like AdaIN. Although InastantStyle-Plus achieves a clearer fusion of color and texture, it still fails to capture the stylistic knowledge of the reference images, tending towards oil painting. StyleID, while performing better than InastantStyle-Plus, only completes simple texture fusion, transforming the image into a colored sketch style without learning the stylistic characteristics of the reference image, such as abstract structural expression. In contrast, *StyleWallfacer* does not have the aforementioned shortcomings. It not only remains faithful to the structural expression of the content image but also to the stylistic expression of the style image, achieving a balance between the two. Additionally, by adjusting model parameters, it can achieve more abstract style transfer results (at which point the style transfer results will be completely faithful to the artistic image's stylistic representation).

### H.4 One-shot Text-driven Style Transfer & Color Edit

The right half of Figure 27 shows the experimental results of one-shot text-driven style transfer & color edit. As depicted, whether on the Watercolor Dogs dataset or the Landscapes dataset, the model is capable of achieving high-quality image color control while completing the style transfer task. This fully demonstrates the model's high degree of customization and controllability during style transfer, which is something traditional methods cannot achieve. This breakthrough expands the boundaries of imagination for style transfer, offering more possibilities for image style transformation.

### H.5 Cross-Content Image Testing Results

To conduct a more comprehensive test of *StyleWallfacer*, we evaluated the model across different image contents. The primary aim of this test was to prevent the same image content from affecting the comprehensiveness of the model evaluation. Therefore, we tested models trained with single images from the Landscape dataset (content of colored pencil sketches of scenery) and the Watercolor Dogs dataset (content of watercolor puppies) using images of "houses". We also tested a model trained with a single image from the Van Gogh Houses dataset (content of Van Gogh-style houses) using images of "gardens". This test fully demonstrates the generalizability of *StyleWallfacer* across images with different contents.

As shown in Figure 28, these are the test results of StyleWallfacer on different image contents. From the figure, it can be analyzed that the model has good generalizability across different image contents, and the generated results excellently achieve a balance between stylization and content preservation.

## I  Limitation and Future Work

In this work, we propose *StyleWallfacer*, a novel unified framework for style transfer. Although the theory presented in this paper has been validated through experiments and demonstrates many advantages over traditional solutions, there are still the following issues:

- While the present study is primarily concerned with the task of infusing stylistic knowledge to achieve a range of style transfer-related objectives, it is imperative to extend the contemplation beyond the confines of this research. Specifically, future endeavors should consider how to leverage the theoretical framework established herein to address tasks analogous to subject-driven image generation. Although *StyleWallfacer* demonstrates remarkable proficiency in incorporating abstract stylistic concepts into models, it appears to be less effective in the context of subject-driven tasks. Consequently, future research may benefit from further exploring strategies for integrating subject-specific knowledge within this domain.

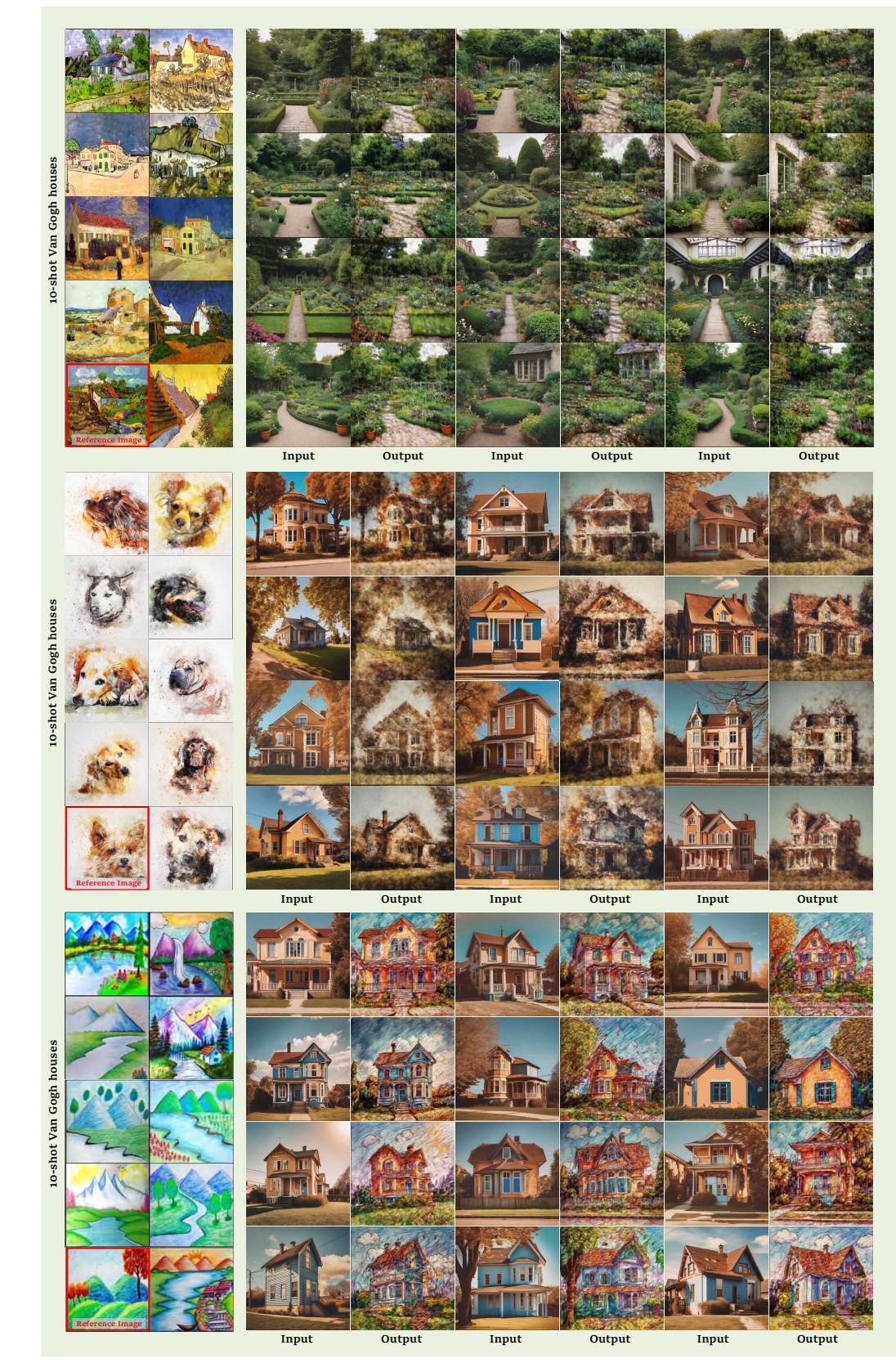

Figure 28: **Qualitative Comparison of Image-driven Style Transfer Results on Different Image Contents.**

- From a theoretical perspective, this paper reveals a significant discrepancy between the natural language processed by text-to-image (T2I) models and that interpreted by humans. However, this study does not fully close this gap. Instead, it integrates the semantic space into the CLIP space as a solution. In fact, future research should focus on how to make the natural language understood by T2I models more consistent with human comprehension, thereby eliminating the need for a "translation" process.

## J  BROADER IMPACT

As an innovative approach to style-driven image generation and editing, *StyleWallfacer* holds potential for application in creative AI endeavors and as a non-traditional form of data augmentation for various downstream tasks. Given that image generation constitutes a fundamental task within the realm of computer vision, the principles underlying *StyleWallfacer* could potentially be extended to research in other related areas. However, akin to other image generation methods, our technique may inadvertently facilitate societal harms such as the creation of counterfeit images for malicious purposes or copyright infringement, contingent upon the specific context of its application. Consequently, we advocate for the judicious and responsible utilization of *StyleWallfacer*.

