# OpenReview forum: "Break Stylistic Sophon: Are We Really Meant to Confine the Imagination in Style Transfer?"
_ICLR.cc/2026/Conference — ICLR 2026 Conference Withdrawn Submission_

### Official Review · Reviewer_evmJ · 2025-10-28

**Soundness:** 1
**Presentation:** 1
**Contribution:** 1
**Rating:** 2
**Confidence:** 5

**Summary:**

This paper proposes StyleWallfacer, a unified style transfer framework addressing semantic drift, overfitting, and color limits via semantic gap–based style injection, human-feedback progressive augmentation, and a training-free triple diffusion process enabling controllable text/image-driven stylization and color editing, outperforming prior one-shot methods.

**Strengths:**

1.This paper is well-written, easy to follow.

2.The style results look good.

**Weaknesses:**

1.This paper claims that StyleWallfacer is the first text-based color editing method; however, to the best of my knowledge, StyleStudio [1] had already introduced a text-driven framework that enables users to edit color attributes using natural language prompts.

2.The descriptions in Introduction about "The three-body problem" and "Wallfacer Plan" are confused and entirely unrelated to the method proposed in this paper.

3.The proposed framework is basically a combination of existing methods. For example, Style LoRA is widely used in AIGC community and decoupled content prompts are explored in StyleShot[2] and StyleCrafter[3]. Attention-based Style/Text control also used in StyleAligned[4] and AlignedGen[5]. Human feedback data augmentation is used in StyleDrop[6]. This paper produces limited insight to style transfer.

4.The citation command \cite was incorrectly. \citet (or \cite) is used when the author’s name is part of the sentence — the citation is integrated into the text (e.g., Smith (2020) proposed...). \citep is used when the citation is parenthetical, meaning it appears in brackets as supplementary information (e.g., ...as shown in previous work (Smith, 2020).).

5.Most of the style references in this paper are paintings, lacking higher-level styles such as 3D, line art, and pixel art.

6.Many state-of-the-art (SOTA) style transfer methods, such as AlignedGen[5], Attention Distillation [7], and OmniStyle [8], are not included in the comparison.

[1]StyleStudio: Text-Driven Style Transfer with Selective Control of Style Elements, CVPR.

[2]StyleShot:StyleShot: A SnapShot on Any Style, TPAMI.

[3]StyleCrafter: Enhancing Stylized Text-to-Video Generation with Style Adapter, TOG.

[4]Style Aligned Image Generation via Shared Attention, CVPR.

[5]AlignedGen: Aligning Style Across Generated Images, NIPS.

[6]StyleDrop: Text-to-Image Generation in Any Style, NIPS.

[7]Attention Distillation: A Unified Approach to Visual Characteristics Transfer, CVPR.

[8]OmniStyle: Filtering High Quality Style Transfer Data at Scale, CVPR.

**Questions:**

See Weakness.

---

### Official Review · Reviewer_XG4e · 2025-10-28

**Soundness:** 2
**Presentation:** 2
**Contribution:** 1
**Rating:** 2
**Confidence:** 4

**Summary:**

This paper introduces StyleWallfacer, a unified framework that tackles key challenges in style transfer by leveraging semantic-driven style injection, progressive data augmentation, and a training-free triple diffusion process to achieve high-quality, diverse stylization with preserved content fidelity.

**Strengths:**

1. Semantic-Driven Style Injection: A novel method using BLIP and LLMs to create and exploit a semantic gap in CLIP space, enabling precise, drift-free style knowledge injection.

2. Progressive Learning via Human Feedback: An innovative data augmentation strategy that iteratively incorporates high-quality generated samples to reduce overfitting and enhance learning.

3. Training-Free Triple Diffusion Process: A clever inference mechanism that manipulates self-attention features to seamlessly blend style and content while maintaining textual controllability, unifying image-driven and text-driven stylization.

**Weaknesses:**

1. Limited Novelty: The core contribution of this paper is somewhat incremental. The claim of being "the first" to achieve color editing during style transfer is overstated, as numerous existing style transfer methods already offer text-guided local or global color control.

2. Incomplete Experimental Comparisons: The chosen baseline methods are not state-of-the-art. To properly validate the proposed method's advantages, comparisons against more recent and advanced techniques are necessary.

3. Factual Inaccuracy in Illustration: There appears to be an error in Figure 2(b), where the text prompt "a face" seems misplaced and should likely be "a church" to match the content of the image.

4. Non-Standard Terminology: The use of "HFRL" is unconventional in the community where "RLHF" (Reinforcement Learning from Human Feedback) is the prevalent and standardized term. This non-standard notation may cause confusion for readers.

**Questions:**

1. In the style knowledge injection phase, could you clarify the comparative performance between training a Style LoRA on a pre-collected set of style images versus your iterative collection approach? Which method proves more effective, and was an ablation study conducted to determine this?

2. Given that the proposed method requires inference through three diffusion models, what is its computational efficiency? Could you provide a direct comparison with other methods in terms of both output quality and efficiency?

3. The choice of using "HFRL" instead of the more common "RLHF" is notable. Could you please explain the rationale behind this terminological decision?

---

### Official Review · Reviewer_nmHW · 2025-10-28

**Soundness:** 2
**Presentation:** 2
**Contribution:** 2
**Rating:** 4
**Confidence:** 4

**Summary:**

This paper proposes a unified style transfer framework called StyleWallfacer which supports both image-driven and text-driven style transfer. StyleWallfacer consists of three key components: a style knowledge injection method based on semantic differences, a progressive learning method based on human feedback for few-shot datasets, and a training-free triple diffusion process that manipulates the features of self-attention layers in a manner similar to the cross-attention mechanism.

**Strengths:**

1. The proposed method supports both image-driven and text-driven style transfer.
2. This paper aims to address the issues present in existing style transfer methods, such as semantic drift, overfitting, and color limitations, which are both meaningful and challenging.
3. Extensive experiments are conducted to evaluate the performance of the proposed method.

**Weaknesses:**

1. This paper is not well-organized and lacks clear presentation: i) The abstract and sections like the introduction in this paper are too lengthy and could benefit from being more concise; ii) The main paper lacks a 'Related Work' section, which makes it harder for readers to understand the developments in the style transfer field; iii) The images presented in this paper have some obvious issues. On one hand, they are too small to be clearly seen; on the other hand, the colorful backgrounds added to the figures blend with inside images, making it hard to distinguish the details.

2. The novelty of the proposed method is limited. (i) The proposed data augmentation strategy based on human feedback (which incorporates high-quality samples generated early in the fine-tuning process into the training set to facilitate progressive learning and significantly reduce its overfitting) has been explored in many previous methods, such as StyleDrop [1]. (ii) The proposed training-free triple diffusion process (which replaces the key and value of the content-related process with those of the style-related process to inject style while maintaining text control over the model) share similar key ideas with StyleID. \
[1] StyleDrop: Text-to-Image Generation in Any Style. NeurIPS 2023.

3. The baselines selected in this paper are not the most state-of-the-art methods currently available. For example, in the text-driven style transfer task, this paper adopted DreamBooth (2023), LoRA version of DreamBooth, Textual Inversion (2023), and SVDiff (2023) as baselines. However, more state-of-the-art methods such as DEADiff (2024) [2], StyleShot (2025) [3], ArtAdapter (2024) [4], StyleStudio (2025) [5] are all neglected. Similar issues exist in other tasks as well. \
[2] DEADiff: An Efficient Stylization Diffusion Model with Disentangled Representations. CVPR 2024. \
[3] Styleshot: A snapshot on any style. TPAMI 2025. \
[4] ArtAdapter: Text-to-Image Style Transfer using Multi-Level Style Encoder and Explicit Adaptation. CVPR 2024. \
[5] StyleStudio: Text-Driven Style Transfer with Selective Control of Style Elements. CVPR 2025.

4. In Figures 7 and 8, the method proposed in this paper does not show a clear advantage over other methods.

5. The paper should be self-contained. It is recommended to carefully plan the length of each section and avoid placing important ablation study results in the supplementary materials; instead, they should be included in the main paper.

**Questions:**

Please see **Weaknesses**.

Others:

What is the time efficiency of different methods? Some evaluations regarding this should be conducted.

---

### Official Review · Reviewer_xEMm · 2025-10-30

**Soundness:** 2
**Presentation:** 1
**Contribution:** 2
**Rating:** 2
**Confidence:** 4

**Summary:**

This paper proposes StyleWallfacer, a high-quality style transfer framework. Its core innovation is a training-free, triple diffusion process that leverages a semantically fine-tuned style LoRA, enabling image stylization and text-based stylization while preserving content structure. The author claimed they effectively overcomes key limitations like semantic drift and overfitting, outperforming existing approaches.

**Strengths:**

+ The paper demonstrates a substantial experimental effort, including a detailed exploration of hyperparameters.

+ The methodological design of the triple-diffusion process is complex and represents a significant engineering endeavor.

**Weaknesses:**

- Overclaim of Contribution: The claim of a "unified style transfer framework" is a significant overstatement. Numerous existing works (e.g., [A,B,C]) have already demonstrated frameworks capable of handling both image- and text-guided style transfer within a single model. The paper does not sufficiently differentiate its "unification" from this established literature.

[A] Wang Z, Zhao L, Xing W. Stylediffusion: Controllable disentangled style transfer via diffusion models[C]//Proceedings of the IEEE/CVF international conference on computer vision. 2023: 7677-7689.

[B] Wang Y, Liu R, Lin J, et al. OmniStyle: Filtering High Quality Style Transfer Data at Scale[C]//Proceedings of the Computer Vision and Pattern Recognition Conference. 2025: 7847-7856.

[C] Wang H, Wu P, Rosa K D, et al. Multimodality-guided image style transfer using cross-modal gan inversion[C]//Proceedings of the IEEE/CVF winter conference on applications of computer vision. 2024: 4976-4985.


- Flawed Motivation on Color Consistency: The paper argues that faithful color reproduction in prior one-shot methods is a limitation. However, color consistency is often an explicit user preference and a desired feature in one-shot style transfer.

- Unsubstantiated Computational Efficiency: The proposed triple-diffusion framework, reliant on multiple parallel denoising paths and inversion techniques, is inherently computationally expensive and memory-intensive. The authors fail to report time consumption and memory usage compared to baseline methods, making it impossible to assess the practical cost of their reported performance gains.

- Impractical and Unjustified Human-Feedback Loop: The introduction of a Human-Feedback Reinforcement Learning (HFRL) stage requiring manual selection of up to 100 samples is highly impractical for real-world style transfer applications. A significant portion of the experiments is conducted under this setting, yet the authors provide no solid ablation studies to demonstrate the critical necessity of this burdensome step for achieving competitive performance.

- Poor Presentation and Readability: The paper lacks a clear, high-level pipeline diagram, making the overall workflow difficult to follow. The quality of the figures is exceptionally poor and visually unprofessional, which severely hinders the ability to evaluate the claimed results.

**Questions:**

Please address the concerns raised in the weaknesses.

---

### Note · Authors · 2025-11-21

I have read and agree with the venue's withdrawal policy on behalf of myself and my co-authors.